# Inductively shunted transmons exhibit noise insensitive plasmon states and a fluxon decay exceeding 3 hours

F. Hassani [1] ✉, M. Peruzzo[1], L. N. Kapoor[1], A. Trioni[1], M. Zemlicka[1] & J. M. Fink [1] ✉

Currently available quantum processors are dominated by noise, which severely limits their applicability and motivates the search for new physical qubit encodings. In this work, we introduce the inductively shunted transmon, a weakly flux-tunable superconducting qubit that offers charge offset protection for all levels and a 20-fold reduction in flux dispersion compared to the state-of-the-art resulting in a constant coherence over a full flux quantum. The parabolic confinement provided by the inductive shunt as well as the linearity of the geometric superinductor facilitates a high-power readout that resolves quantum jumps with a fidelity and QND-ness of >90% and without the need for a Josephson parametric amplifier. Moreover, the device reveals quantum tunneling physics between the two prepared fluxon ground states with a measured average decay time of up to 3.5 h. In the future, fast time-domain control of the transition matrix elements could offer a new path forward to also achieve full qubit control in the decay-protected fluxon basis.

Since the first observation of coherent Rabi oscillations two decades ago[1,2], superconducting qubit coherence times improved by several orders of magnitude[3]—recently reaching the millisecond mark[4,5]. The community owes this success to continuous parallel innovations and effort put into improving the fabrication process[5–7], more thorough shielding and isolation from the environment[8], but also to a much-improved understanding and control of the circuit sensitivities to various noise sources.

Controlling the circuit potential and the resulting variance of the qubit state wavefunctions provides an essential tool for reducing the dispersion of the qubit levels and for engineering noise-protected states[9]. This strategy was particularly successful in the case of the transmon qubit, a charge qubit that operates in the limit of large Josephson to charging energy ratio $E_J/E_C \gg 1$[10,11], thus delocalizing the qubit wavefunctions over multiple charge basis states and flattening its charge dispersion. More recently, this was also achieved in the case of rf-SQUID type qubits by realizing quasi-charge qubits that operate in the challenging to realize high impedance, i.e., low inductive to charging energy ratio $E_L/E_C \ll 1$[12,13], thus delocalizing the wavefunction in phase and flattening the flux dispersion.

In this work, we present a different strategy to achieve the latter, i.e., the comparably easy to realize—but so far unexplored—limit of $E_J/E_C, E_J/E_L \gg 1$ as shown in Fig. 1. This inductively shunted transmon (IST) limit does not rely on particularly high impedance but rather on making use of plasmon levels—a characteristic of charge qubits—in an rf-SQUID qubit geometry that traditionally relies on flux encoding. There are a number of proposals that introduce a variant of such a qubit as a suitable device to implement longitudinal coupling[14,15] or to explore non-abelian many-body states[16]. More moderate parameter regimes are being explored to optimize the transmon toward higher anharmonicity, controlled flux tunability and resulting higher gate fidelities[17].

Even though the plasmon qubit encoding in the deep IST limit studied here shares many similarities with the transmon—including its eigenenergy, anharmonicity and transition matrix elements—there are also a number of important differences. The large inductive shunt of the IST decompactifies the phase of the transmon[18], meaning that it localizes continuous qubit wavefunctions in wells with discrete flux number.

[1]Institute of Science and Technology Austria, 3400 Klosterneuburg, Austria. ✉e-mail: farid.hassani@ist.ac.at; jfink@ist.ac.at

The inductor therefore leads to quadratic confinement of the qubit wavefunction in the phase variable and can potentially stabilize the average number of qubit excitations in the presence of large photon number in the cavity[19,20]. Such dynamical instabilities[21-23] are believed to prevent quantum-non-demolition (QND) qubit readout with high probe powers and might also put a limit on the generation of even larger photon number dissipative cat-qubits with improved $T_1$ protection[24].

After going over the theory of the IST qubit and its relationship to the fluxonium and transmon, we present the device design, spectroscopy and time-domain coherence measurements for 3 devices with different $E_L$. We furthermore show that the established tools from circuit QED, i.e., the spectroscopically determined plasmon transition frequencies, can be used to read out the long-lived local fluxon ground states, which reveal interesting tunneling physics away from zero flux and at elevated temperatures. Then we continue by discussing the fidelity and QND-ness of the high-photon number readout of the IST plasmon states. Finally, we show deterministic fluxon state preparation and direct time-domain measurements of the fluxon decay rates.

## Results

### Theory

The Hamiltonian of the IST qubit is that of the rf-SQUID shown in the inset of Fig. 2a and given as

$$H = 4E_C \hat{n}^2 - E_J \cos(\hat{\phi}) + \tfrac{1}{2}E_L(\hat{\phi} + \varphi_{\text{ext}})^2, \qquad (1)$$

where the first two terms describe a regular transmon qubit with the two canonical variables charge $\hat{n}$ and phase $\hat{\phi}$. Adding the inductive energy term adds a quadratic confinement in the flux degree of freedom and lifts the periodicity of the $\cos(\hat{\phi})$ potential, which enables flux transitions between neighboring wells. The spectrum shown in Fig. 2a, unlike the transmon, is invariant to charge offset[25] and is a function of the external magnetic flux $\varphi_{\text{ext}} = 2\pi\Phi_{\text{ext}}/\Phi_0$ instead.

At zero external flux $\varphi_{\text{ext}} = 0$, the first transition between the ground and first excited state is located within one well, as shown in Fig. 2b and approximately given by the plasmon frequency $\omega_p = \sqrt{8E_J E_C}$. In this limit, the plasmon transition energies are mostly a function of the shape of the bottom of the single cosine well and, therefore, comparably insensitive to external flux.

As the magnetic field is tuned to half flux $\varphi_{\text{ext}} = \pi$, the potential changes to a double well configuration shown in Fig. 2c favoring low energy fluxon transitions—the natural basis states of loop-type qubits such as flux and fluxonium qubits[26-28]. In this bias condition long

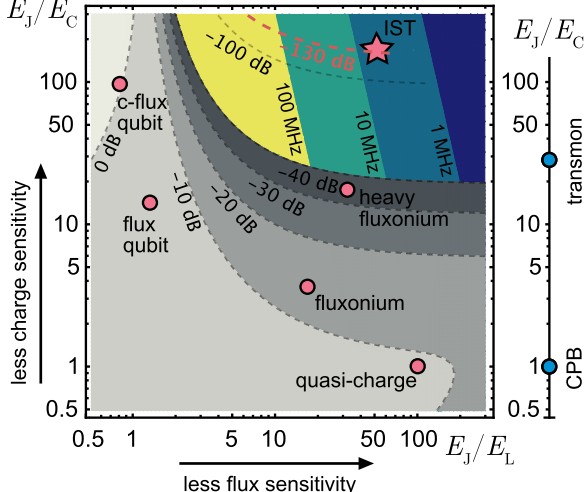

**Fig. 1 | Classification scheme for superconducting qubits.** We use the characteristic energy ratios $E_J/E_L$ and $E_J/E_C$ to parametrize the plot[71-73]. The values for the Cooper pair box (CPB), transmon, flux qubit, capacitively shunted flux qubit (c-flux qubit), fluxonium, heavy fluxonium and quasi-charge qubit are taken from refs. 11, 12, 27, 35, 45, 74 respectively. We note that the quarton and unimon qubit as well as the SNAIL element are close to the flux and c-flux qubit, respectively, in this parametrization[75-77]. The star depicts the parameters of the inductively shunted transmon (IST) qubit (device A listed in Table 1). The gray color scale and dashed gray lines are a contour plot of the matrix element of the lowest fluxon transition calculated at half flux $\varphi_{\text{ext}} = \pi$ on a logarithmic scale for the fixed Josephson energy $E_J/h = 29.9$ GHz of device A. The color-coded contour areas in the fluxon transition suppressed region show the flux dispersion of the plasmon state over a full flux quantum as calculated from Eq. (2) for the same $E_J$.

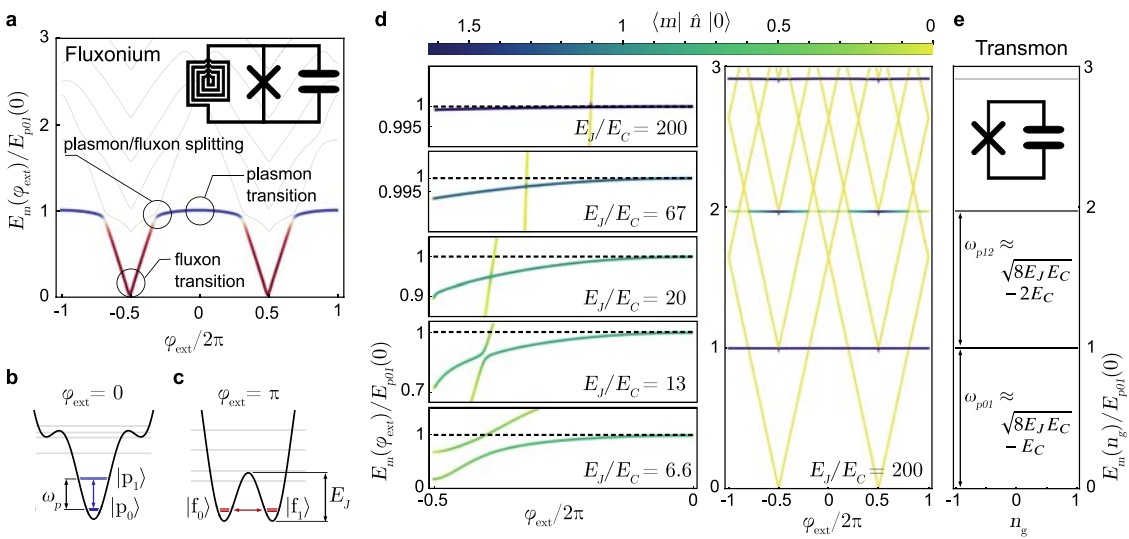

**Fig. 2 | Evolution from the fluxonium to the inductively shunted transmon spectrum. a** Circuit (inset) and spectrum of a typical fluxonium device with $E_J/h = 3$ GHz, $E_L/h = 0.5$ GHz and $E_C/h = 0.45$ GHz[47] as a function of external flux. The red and blue colors indicate the fluxon and plasmon transition of the ground to the first excited state, respectively. **b** and **c** show the potential for plasmon and fluxon transitions at zero and half flux, respectively. **d** First column shows the transformation of the fluxonium to the IST qubit spectrum by increasing the $E_J/E_C$ ratio ($E_C/h = 150$ MHz and $E_L/h = 500$ MHz), where the color scale indicates the calculated matrix elements. The second column shows the full spectrum of the IST qubit, including the diamond-shaped flux levels with ultra-small matrix elements. The low dispersion plasmon levels are in agreement with the transmon spectrum shown in panel (**e**) for the same $E_J$ and $E_C$.

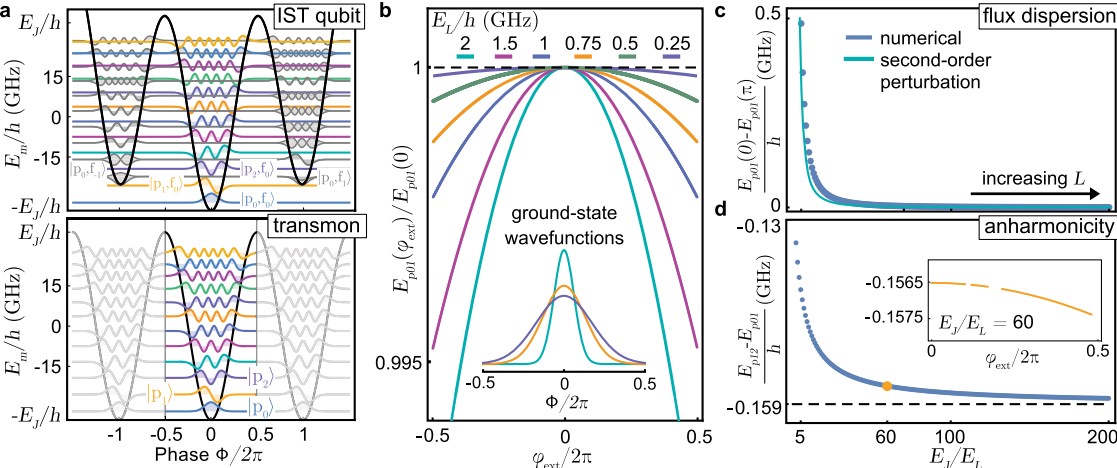

**Fig. 3 | Quantitative comparison of the IST and transmon qubits. a** The potential (black line), wavefunctions (colored and gray lines) and eigenenergies (y-axis offsets) of the IST qubit (top) for $\varphi_{ext} = 0$ and the equivalent transmon without inductive shunt (bottom). The high tunneling barrier given by $E_J/h = 35$ GHz, along with the heavy mass ($E_C/h = 0.15$ GHz), renders the quadratic confinement of the wells given by $E_L/h = 0.5$ GHz of the IST to be a mere perturbation for the lowest energy levels. This leads to plasmon wavefunctions (colored) resembling closely those of the transmon (bottom). The gray wavefunctions in the top panel show the practically inaccessible flux transitions for (exactly) $\varphi_{ext} = 0$. **b** Lowest energy plasmon transition (colored lines) as a function of $\varphi_{ext}$ for the same fixed $E_J$ and $E_C$. The choice of $E_L$ determines the amount of phase confinement. Higher inductance leads to weaker phase confinement, a larger ground state wavefunction variance (inset) and a strongly reduced flux dependence. **c** The calculated full flux dispersion as a function of $E_J/E_L$ for the same fixed $E_J$ and $E_C$. The results from numerical diagonalization (blue points obtained with ref. 39) agree with the quadratic suppression predicted by Eq. (2) based on second-order perturbation theory (green line). **d** The calculated qubit anharmonicity as a function of $E_J/E_L$ for the same fixed $E_J$ and $E_C$. In the high $E_J/E_L$ limit, the anharmonicity of the IST (blue points) converges to that of the transmon (dashed line). The inset shows that the change of this anharmonicity as a function of external magnetic flux is very small for large values of $E_J/E_L$. The discontinuities are the result of fluxon states crossing with the plasmon level where the numerical algorithm fails to follow the plasmon state reliably.

energy relaxation times[29] as well as protection from quasi-particle loss[30] have been demonstrated.

At intermediate flux values, the spectrum in Fig. 2a exhibits a plasmon/fluxon transition splitting. In the high inductance limit $E_J/E_L \gg 1$ the size of this splitting is a measure of the inter-well coupling, as shown in Fig. 2d. In the low-capacitance (light) regime characterized by small $E_J/E_C \sim 1$, the splitting is very large. In the case of ultra-high impedance, the splitting opens up and forms flat Bloch bands that form the basis states of the recently realized quasi-charge qubit[12,13]. In the high-capacitance (heavy) regime characterized by $E_J/E_C \gg 1$, on the other hand, the circuit is characterized by plasmon transitions.

The diamond-shaped fluxon transitions in Fig. 2d are exponentially suppressed, with the matrix element calculated to be as low as $10^{-13}$ due to the heavy nature of device A with $E_J/E_C = 182$. Here the plasmon/fluxon splitting is closed, and the plasmon levels form flat bands with extremely small flux dispersion on the order of a few MHz, cf. Fig. 2d and dashed contour lines in Fig. 1. In Figure 2e, we show that in this regime, the IST plasmon transition energies are in good agreement with those of the equivalent transmon circuit without the large inductive shunt - except that those bands are flat with respect to the gate charge $n_g$ rather than $\varphi_{ext}$.

In Fig. 3a, we compare the potential, eigenenergies and wavefunctions with those of the transmon to acquire more intuition about the properties and spectrum of the IST qubit. While the transmon potential and wavefunctions extend periodically from minus infinity to infinity, the lowest energy IST wavefunctions are localized in one specific well. Because the inductive confinement lifts the neighboring wells by a small energy compared to the depth of the wells given by $E_J$, the shape of the potential, the resulting plasmon wavefunctions, and eigenenergies resemble very closely those of the transmon. The matrix elements between flux states (gray wavefunctions) are exponentially suppressed by the very large barrier between the wells (compared to the plasmon energy).

In the small $E_L$ (large $L$) limit, the shape of the well is mostly determined by the Josephson energy, and the lowest plasmon

transition energy therefore becomes extremely flat with respect to flux, as shown in Fig. 3b, exhibiting a relative flux dispersion of less than one part in a thousand. As expected, this insensitivity is accompanied by an increased variance of the ground state wavefunction in phase, as shown in the inset. The IST qubit therefore realizes flux noise insensitivity by increasing $E_J/E_L$ in analogy to the charge noise insensitivity of the transmon obtained for large $E_J/E_C$.

To get more insight into the scaling of the flux noise protection, we analyze the Hamiltonian Eq. (1) under the assumption that the inductive part of the Hamiltonian acts as a local perturbation ($E_L \ll E_J$). Supplementary Note 1 covers the derivation and a comparison to numerical results. In the limit where $E_J/E_C \gg 1$, we obtain a simple expression for the flux dispersion of the first plasmon qubit transition

$$\frac{\partial \omega_{p01}}{\partial \varphi_{ext}} \approx -\frac{\sqrt{8 E_J E_C}}{\hbar (2 E_J/E_L)^2} \varphi_{ext}. \qquad (2)$$

It shows that the $E_J/E_L$ ratio provides a quadratic suppression of both the first- and second-order derivatives, which are the relevant quantities for qubit dephasing at intermediate and zero flux[31]. Furthermore, Eq. (2) also shows that, in analogy to the Cooper pair box where the transition frequency is a function of charge offset squared $\omega_{01}(n_g^2)$, the IST qubit transition is also given by a parabola but versus external flux $\omega_{p01}(\Phi_{ext}^2)$.

Figure 3c shows the full dispersion $E_{p01}(0) - E_{p01}(\pi)$, calculated with Eq. (2) for a fixed set of $E_J$ and $E_C$. The quadratic prediction (green line) matches very well with the numerical results (blue points) for a large range of $E_J/E_L$. Figure 3d shows the calculated anharmonicity $E_{p12} - E_{p01}$ of the qubit for the same parameters with $\omega_{p01} \approx 6$ GHz as a function of the $E_J/E_L$ ratio. As the inductance of the superinductor increases, the Hamiltonian in Eq. (1) converges to that of the transmon, and therefore the anharmonicity of the qubit also converges to the transmon anharmonicity (dashed line). However, for low inductance, the parabolic potential dominates, which results in lower anharmonicity.

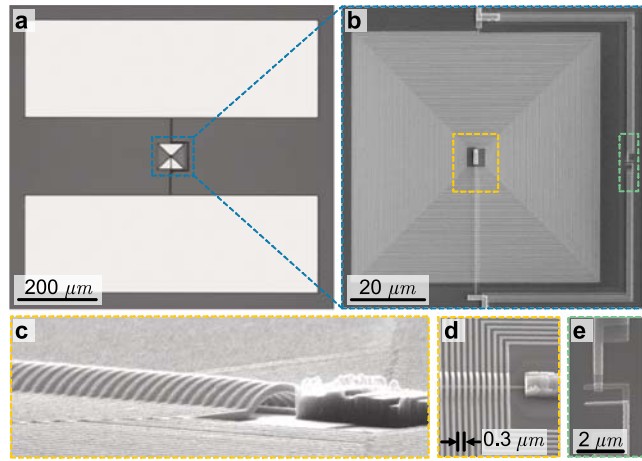

**Fig. 4 | IST qubit scanning electron microscope images. a** Overview image of the aluminum capacitor pads (white) fabricated with an inductively coupled plasma etching recipe on high resistivity silicon (dark gray). **b** Enlarged view of the aluminum geometric superinductor (-14 mm long) and Josephson junction shunting the qubit capacitor. **c** Isometric view of the central airbridge part of the inductor and the patch layer, which is deposited after ion gun etching to ensure a reliable electrical contact between the coil, capacitor and junction aluminum layers. **d** Enlarged view of the center of the coil with 99 turns and a wire width and spacing of 150 nm, respectively. It is fabricated using an inductively coupled plasma etching recipe and device C from the first generation exhibits a few shorts which leads to a three times lower inductance. **e** Enlarged view of the Josephson junction fabricated with the Dolan bridge method[78].

### Table 1 | Extracted qubit parameters

|   | $E_J/h$ (GHz) | $E_C/h$ (GHz) | $E_L/h$ (GHz) | $g_0/(2\pi)$ (MHz) | $T_1$ (µs) | $T_2$ (µs) | $\nu_{p01}$ (GHz) | $\delta\nu_{p01}$ (MHz) |
|---|---|---|---|---|---|---|---|---|
| **A** | 29.93 | 0.164 | 0.56 | 107.7 | 15.5 | 13.0 | 6.122 | 5.1 |
| **B** | 31.13 | 0.165 | 0.56 | 119.6 | 21.0 | 27.8 | 6.296 | 5.6 |
| **C** | 33.34 | 0.170 | 1.60 | 86.3 | 17.4 | 22.6 | 6.720 | 40.0 |

The reported coherence times are measured at $\varphi_{ext} = 0$. Echo experiments on device C did not improve the $T_2$ time, which suggests that low-frequency flux noise is not the dominant limitation at zero flux. $\delta\nu_{p01}$ refers to the measured qubit dispersion over the full flux range.

### IST qubit devices

The three studied IST qubit devices are based on the 3D transmon design[32] with a single Josephson junction with $E_J/h \approx 30$ GHz and a shunting capacitance of $C_s \approx 100$ fF, as shown in Fig. 4. The large inductor shunting the Josephson junction is based on a miniaturized planar coil[33] with a large inductance of 100–300 nH. The effective qubit parameters are listed in Table 1 and the fabrication details are to be found in Supplementary Note 2.

The fabricated devices are packaged in a rectangular 3D cavity made from oxygen-free copper with the first resonance mode $\nu_r \approx 10.48$ GHz, an internal quality factor of $Q_i = 2.7 \times 10^4$ and a total loss rate of $\kappa/2\pi \approx 1$ MHz. The cavity is then attached to the cold plate of a dilution refrigerator at a temperature of 7 mK. The qubit is controlled and read out via the cavity port using microwave pulses passing through multiple stages of attenuation, a 12 GHz lowpass filter, an Eccosorb filter, and finally, a circulator to reach the cavity. The qubit readout is done based on the reflected signal that passes through two stages of isolators, an 8–12 GHz bandpass filter, a low-noise high electron mobility amplifier at the 3 K stage followed by another low-noise amplifier and demodulation at room temperature. We use a large radiation shield that is coated with a mixture of Stycast and carbon powder and thermalized in the mixing chamber. Inside it, the cavity is located on the bottom part of a double-layer cryogenic $\mu$-metal shield to minimize stray magnetic fields (see Supplementary Fig. 3).

### IST qubit spectroscopy

We use dispersive readout and two-tone spectroscopy[34] to obtain the qubit spectrum shown in Fig. 5a. Surprisingly, the spectrum at the base temperature of the dilution refrigerator does not show a periodic behavior with external flux as predicted by Eq. (1). Flux periodicity is a crucial feature of any flux-tunable device as it provides the unit-less flux scaling and, in turn, allows to infer the qubit energies. In usual rf-SQUID devices—including very heavy fluxonium qubits[35–38]—the global ground state of the system switches from one well to a neighboring well of the potential landscape at $\varphi_{ext} = \pi$ as a function of external flux, and the fictitious phase particle always moves to the ground state well due to the non-negligible inter-well coupling. However, in the case of the IST qubit circuit, the phase particle stays trapped within its local minimum due to the high barrier formed by the large Josephson energy and the heavy mass (large shunt capacitance) of the phase particle. Only when the local minimum exceeds a critical value, which in our case is more than one $\Phi_0$ in flux bias or ≈15 GHz in energy from the ground state, a probabilistic tunneling event is triggered by vacuum or thermal fluctuations as shown in Fig. 5a (top). This frequency difference is approximated using $2\pi\Delta\varphi_{ext}E_L/h$, where $\Delta\varphi_{ext}$ corresponds to the difference between $\varphi_{ext}$ at which tunneling occurs and the half flux degeneracy point. At the base temperature, we do not observe any such switching events on the time scale of hours for bias values close to $\varphi_{ext} = \pi$. This is an extreme case for the expected $T_1$ protection of the flux states in this limit[36] which we explore in detail in the final subsection.

In order to regain the flux periodicity of the spectrum of Eq. (1), we controllably increase the temperature of the device with a heater on the mixing chamber plate of the dilution refrigerator. The spectrum is stable without additional tunneling events up to around 100 mK above which we observe a drastic increase in the number of switching events, as shown in Fig. 5a. These random events add up to a consistent, smooth and periodic flux dependence at 250 mK, as shown in Fig. 5b. This measurement probes the plasmon spectrum starting from a thermal mixture of all accessible qubit states where each parabola represents the first plasmon transition of an individual well in the circuit potential. Point **A** in Fig. 5b shows the flux sweet spot where one specific well is located at its minimum energy. Point **B**, on the other hand, indicates the degeneracy point between two neighboring wells, and point **C** shows a second-order degeneracy between two next-neighbor wells. The amplitude of any parabola beyond half flux bias gradually vanishes, which indicates that the probability of finding the system in the higher energy neighboring well is significantly lower than finding it in its global minimum well.

Importantly, and different from other mechanisms that can induce uncontrolled flux discontinuities, such as when external flux vortices move in the vicinity of the rf-SQUID loop, we can also reconstruct a smooth spectrum from low-temperature data. Combining a set of independent flux sweep measurements (with fixed frequency and external flux range), all conducted at the base temperature of 7 mK, in one plot yields the data shown in Fig. 5c, d for the resonator dispersive shift and the qubit frequency, respectively. Using the periodicity found in Fig. 5b, we solve the Hamiltonian in Eq. (1) numerically using the scQubits python library[39] to obtain the eigenenergies and fit (black lines) the characteristic energies and the qubit-resonator coupling of device B as listed in Table 1. More details about the fitting procedure that also takes into account weak coupling to parasitic modes are found in Supplementary Note 4.

The observed phase tunneling physics and flux frustration highlights that the IST qubit can be considered a close relative of the phase qubit where the very-high linear inductance acts as a current bias for

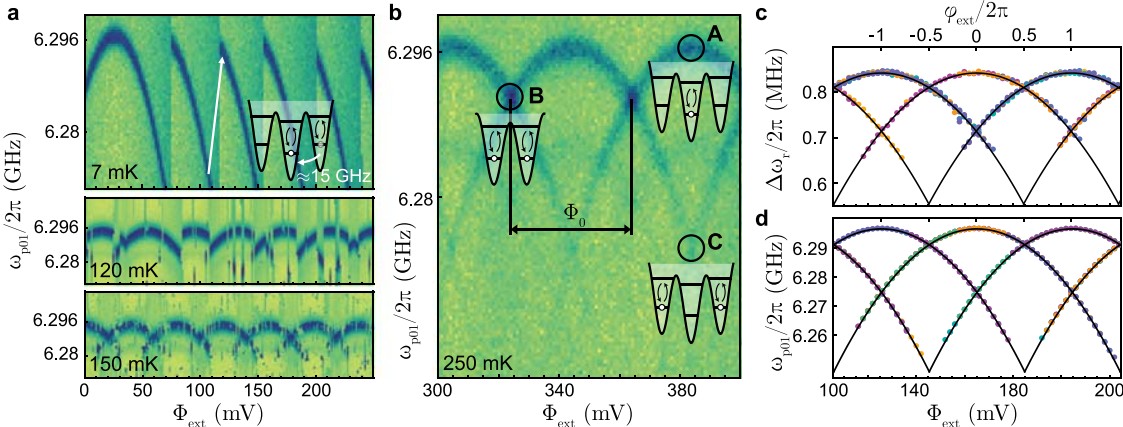

**Fig. 5 | Qubit spectroscopy and macroscopic quantum tunneling. a** Measured spectrum of the lowest energy plasmon mode versus magnetic flux in units of the voltage applied to an external bias coil using device B. At the base temperature, the parabolic qubit spectrum exhibits discontinuities when a certain normalized bias value is exceeded. Each jump to higher frequencies corresponds to a quantum tunneling event of the circuit's fictitious phase particle that remains trapped in a stable flux state up to and beyond a full flux quantum of external flux corresponding to ≈15 GHz above the ground state[66,79,80]. This process is shown in the inset with arrows (not to scale). At higher temperatures, these tunneling events are triggered by thermal fluctuations and become more frequent–also for small bias values. **b** Plasmon level spectroscopy at 250 mK results in a smooth spectrum that contains a thermal mixture of all tunneling events and populations. This allows to identify the magnetic flux quantum $\Phi_0$ and individual parabola that correspond to

specific and distinguishable potential wells with an integer flux $m$ being occupied. Point **A** of the inset labels refers to the plasmon transition sweet spot where $\Phi_{ext} = m\Phi_0$ and, starting from the flux ground state, the highest transition frequency is measured. At the half flux point **B**, the two neighboring wells become degenerate with identical and somewhat lower plasmon transition frequency. Point **C** identifies the degeneracy point between two wells left and right of the global ground state. The spectrum continues periodically outside of the shown range. **c** and **d** show a fit to the readout resonator dispersive shift $\omega_r/2\pi - 10.459$ GHz and plasmon qubit frequency, obtained at base temperature and based on 10 (color-coded) consecutive flux sweep data sets (5 in each direction). The fit (black lines) was obtained by first fixing $E_J$ and $E_C$ using point **A** in panel (**b**), then using point **B** and **C** to obtain the inductive energy $E_L$.

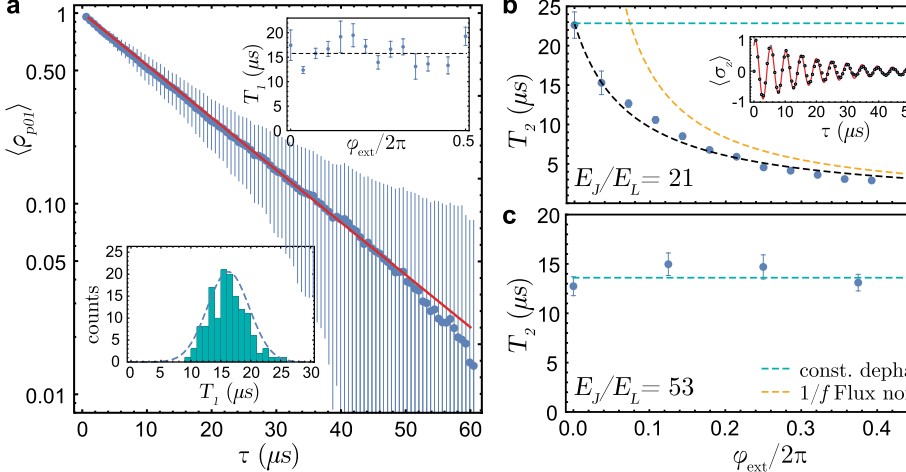

**Fig. 6 | Qubit plasmon relaxation and dephasing measurements. a** The normalized readout voltage proportional to the average excited state population of device C after a $\pi$ pulse excitation of the $|p_{01}\rangle$ state at various flux values. The blue points are the mean of all measured traces, and the bars show the standard deviation of the individual measurements. The data does not show a direct indication for quasi-particle induced loss and fits well to a single decay exponential function (red line) yielding $T_1 = 15.8$ μs. The histogram of all 120 measured relaxation times (bottom inset) agrees with a single-peaked Gaussian envelope (dashed line). The measured $T_1$ times are approximately constant vs. external flux (top inset, error bars show statistical standard error), but the observed fluctuations around the mean (dashed line) suggest two-level-system coupling and dielectric losses.

**b** and **c** show $T_2$ decoherence times obtained from standard Ramsey measurements (top inset) for device C with $E_J/E_L = 21$ and device A with $E_J/E_L = 53$, respectively. The maximum $T_2 = 22.6$ μs of device C is strongly reduced by flux noise away from the integer flux sweet spot. We fit the flux dependence (dashed black line) using the measured mean $T_1$, a thermal photon shot noise in the resonator $T_{th} \approx 90$–100 mK (dashed cyan line) and a $1/f$ flux noise amplitude of $A_\Phi = 98$ μ$\Phi_0$ (dashed yellow line). Device A shown in panel (**c**), on the other hand, exhibits strong dephasing protection due to its large inductance. Over the full flux range, $T_2$ is scattered around the mean $T_2 = 13.6$ μs (dashed cyan line) without a clear flux dependence. It was possible to measure the $T_1$ and $T_2$ data at each flux value for around 30 minutes without unwanted switching events, also for values very close to half flux.

the Josephson junction while preserving the shape of the potential well and suppressing the band dispersion[40]. The observed flux trapping is also related to ref. 41, where the escape of the phase particle is observed in a device formed by two parallel Josephson chains coupled capacitively to a resonator, as well as to the hysteresis observed in rf-SQUID type Josephson parametric amplifiers[42]. Nevertheless, we are

not aware of any realizations of this physics in a superconducting qubit or any other non-distributed single junction device.

## Plasmon qubit coherence

Here we report the time-domain characterization of the plasmon qubit transition. All $T_1$ measurements over the full flux range of device C are

shown in Fig. 6a. The energy relaxation of the $|p1\rangle$ state (as defined in Fig. 3a) is shown on a logarithmic scale, as obtained from 120 individual $T_1$ measurement sweeps equally distributed over the full flux range. We find no sign of a double decay, which would indicate the presence of a relevant amount of quasi-particle-induced loss[43–45] and the histogram reveals a single-peaked normal distribution.

Based on Fermi's golden rule alone[46], we do not expect a $T_1$ dependence on the external flux since the transition frequency and its matrix element stays approximately constant over the entire flux range. Experimentally (Fig. 6a top inset), we observe a random variation around an otherwise constant mean of $T_1 = 15.5\,\mu s$, corresponding to an effective quality factor of $Q_q = 0.67 \times 10^6$, on par with some of the best values in the literature[47]. The relatively high matrix element and transition frequency of the plasmon state render it susceptible to dielectric losses, and the observed variation indicates possible two-level-system coupling[48]. Material and design improvements based on a study of the participation ratios of the electric field distribution[49] and its interaction with the geometric superinductor could potentially overcome this limitation.

In Fig. 6b, c, we compare the effect of flux noise protection for device C with $E_L = 1.6\,GHz$ and device A with $E_L = 0.56\,GHz$ over the full flux range, respectively. Device C shows a significant drop in measured $T_2$ times away from the flux sweet spot, while device A, with the three times higher inductance, exhibits no clear dependence of $T_2$ to external flux over the full flux range.

We model and fit (black dashed lines) the total decoherence rate with $\Gamma_{T_2} = \Gamma_{1/f} + \Gamma_{th} + 1/(2T_1)$, where $\Gamma_{1/f}$ is due to flux noise and $\Gamma_{th}$ due to resonator photon shot noise. Dephasing due to $1/f$ flux noise can be expressed as $\Gamma_{1/f} = \sqrt{\gamma A_\Phi}\,|\frac{\partial \omega_{p01}}{\partial \Phi_{ext}}|$ and using Eq. (2), we obtain

$$T_{1/f} = \frac{1}{\Gamma_{1/f}} = \frac{\hbar \Phi_0 (2E_J/E_L)^2}{4\pi^2 \varphi_{ext} \sqrt{\gamma A_\Phi} \sqrt{8E_J E_C}}, \qquad (3)$$

where $\sqrt{A_\Phi} \approx 98\,\mu\Phi_0$ is the flux noise amplitude and $\gamma = \ln \frac{f_u}{2\pi f_l} \approx 9.9$ represents the scaling parameter for the specific Ramsey sequence noise filter function with low- and high-frequency cutoffs $f_l = 250\,mHz$ (inverse measurement time per data point) and $f_u = 1/T_2 = 31\,kHz$[31,50]. The reported number for the flux noise amplitude is obtained by fitting Eq. (3) to the measured external flux-dependent $T_2$ time shown in Fig. 6b. This flux noise amplitude is found to be larger than the typical values reported in the literature, which we attribute to the large effective loop perimeter created by the geometric superinductance[13,51]. Its contribution to the total dephasing is depicted in Fig. 6b (dashed yellow line).

The flux-independent thermal photon-induced dephasing is calculated according to ref. 52 and shown together with the measured $1/(2T_1)$ limit in Fig. 6b, c (cyan dashed lines). From the fit, we obtain a thermal resonator occupation of $n_{th} = 0.004$ for device C shown in panel b and $n_{th} = 0.011$ for device A shown in panel c. The difference could be explained partly by the fact that the resonator of device A is coupled stronger to external drive and readout line, but we note that its coherence might also be limited by a different flux-independent dephasing mechanism.

The effective dephasing model (black dashed line) agrees well with the measured $T_2$ times of device C shown in Fig. 6b. Devices A and B have the largest inductance and exhibit a much larger ratio $E_J/E_L \approx 53$, which results in a drastically reduced flux dispersion. While the $T_2$ data shown in Fig. 6c fluctuates as a function of flux, we do not observe a systematic reduction of $T_2$ up to $\varphi_{ext} = \pi$. In case of device B (not shown), we observe a larger variation, but the maximum $T_2 \approx 28.5\,\mu s$ is measured at $\varphi_{ext} = \pi/2$. Given the two orders of magnitude higher flux noise amplitude, the result shown in Fig. 6c represents a new level of dephasing protection in comparison with the most coherent flux-tunable qubits[38,47,53] away from the flux sweet-spot.

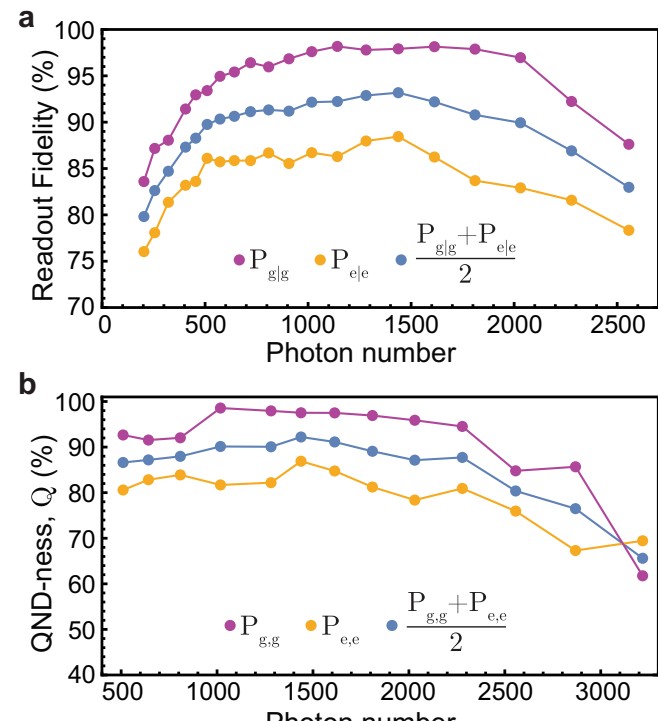

**Fig. 7 | Single-shot readout fidelity and QND-ness. a** Shows the qubit readout fidelity with respect to readout power. Here, the $P_{g|g}$ and $P_{e|e}$ are defined as the probability of preparing ground (excited) state and subsequently also measure the qubit to be in ground (excited) state. Ground and excited state initialization are implemented with a 200 μs waiting time and a 40 ns long gated microwave pulse, respectively. **b** Shows the QND-ness of this measurement as a function of $\bar{n}$. Here, $P_{g,g}$ and $P_{e,e}$ are the probabilities to measure the ground and excited state consecutively in two independent 500ns long measurement pulses that are 1 μs apart, cf. Supplementary Fig. 5a. We note that the reported photon numbers correspond to calibrated steady state powers and do not reach the full value for a short readout pulse.

## High-power QND qubit readout

In this subsection, we study the high photon number resilience of the IST qubit (device C) at zero flux. We observe quantum jumps in continuous measurements, as shown in Supplementary Fig. 5a, and perform high-fidelity single-shot qubit readout. These results are achieved without any kind of parametric amplifier but instead by increasing the power of the measurement tone corresponding to an intra-cavity photon number of $\bar{n} > 1000$ without a significant degradation of the quantum-non-demolition (QND) character of the readout.

We obtain the fidelity of the qubit readout by calculating the probability of measuring the qubit in the excited or ground state conditioned that the qubit was initialized in the excited or ground state respectively, shown as $P_{e|e}$ and $P_{g|g}$ in Fig. 7a. For this, we use a single readout pulse with 500ns length, and we repeat the measurement $40 \times 10^3$ times to collect data points for extracting the quadrature histograms and sweep the measurement power, as shown in Fig. 7b. The highest combined readout fidelity is 93.2% at $\bar{n} \approx 1500$ with a significantly higher fidelity of 98.3% for the ground state. We attribute the lower excited state fidelity of 86.7% to the comparably lower $T_1 \approx 7\,\mu s$ (<7%) compared to previous cooldowns, as well as the expected state preparation errors due to the lack of pulse-shaping the excitation pulse (>5%). Increasing the measurement power further lowers the measured state fidelities. This can be explained by leakage to the ground state of the neighboring well, which manifests as a new maximum in the readout quadrature histogram, as shown in Supplementary Fig. 5d.

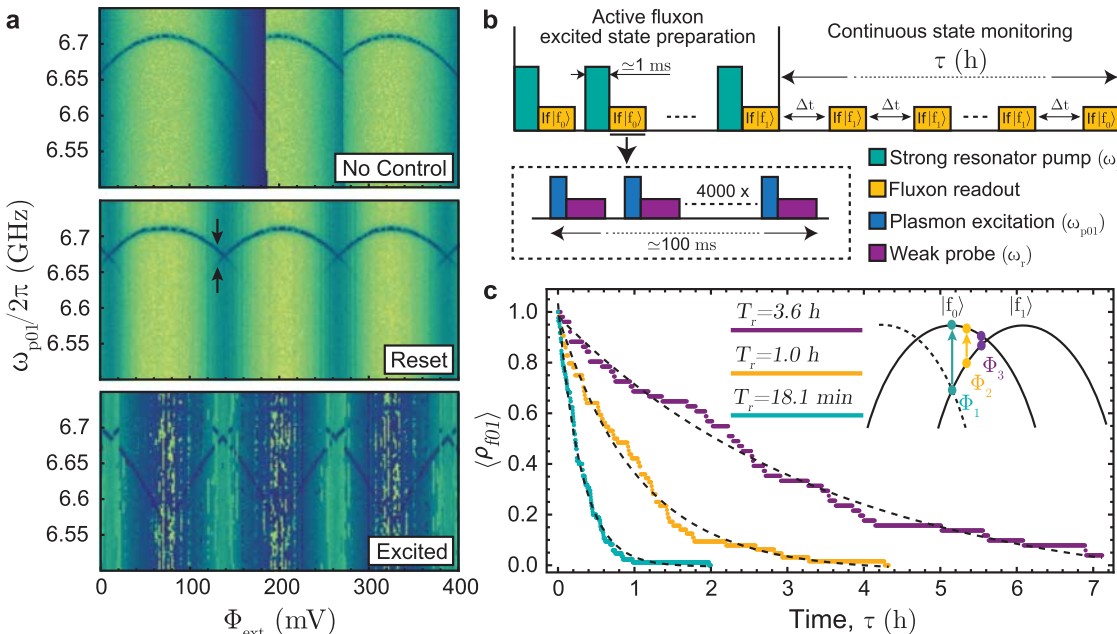

**Fig. 8 | Preparation and lifetime of decay-protected fluxon states. a** Measured spectrum of the lowest energy plasmon mode versus magnetic flux of device C. The top panel was measured without fluxon preparation and shows the non-periodic spectrum similar to Fig. 5 at 7mK for device B. The middle panel was measured right after a 1 ms long resonator pump tone with $\bar{n} = 100 - 2000$. Here the periodicity is restored, and the fictitious phase particle is locked to the global minimum of the potential landscape. Near the flux frustration point, the ground state initialization is not perfect as indicated by two visible lines (black arrows). The spectrum in the bottom panel shows the same measurement with only $\bar{n} = 70 - 80$ resulting in an excited fluxon state, i.e., the fictitious phase particle is generally found in the lowest energy neighboring well rather than the global minimum. **b** Measurement sequence used for the data in panel (**c**). First, we actively determine the fluxon state

(yellow) immediately after the excitation pulse (green) and repeat the sequence until the fluxon excitation is successful. Then, we continue to measure the state every $\Delta t = 30\,\text{s}$ until it has decayed via a quantum tunneling event. The fluxon readout (yellow) is based on the fluxon-dependent plasmon frequency away from half flux. Applying a plasmon $\pi$ pulse (blue) is only successful if the plasmon level is on resonance, i.e., in a certain flux well. The resulting plasmon excitation is measured via the standard dispersive readout for $15\,\mu\text{s} \approx T_1$ (purple) in the single photon limit to avoid photon-induced fluxon tunneling and repeated 4000 times for high SNR. **c** Fluxon decay from the excited to the ground state measured for three different flux bias configurations (dots) and exponential fits (lines). The data is based on $\approx 80$ decay events each, conducted over up to 1 week of raw measurement time for the longest decay time.

Measuring with a high photon number probe tone leads to a high signal-to-noise ratio (SNR) but is typically associated with a destructive readout. The degree of the quantum-non-demolition character of the measurement can be obtained by extracting the correlations between two consecutive measurements of the prepared qubit state[54–56]. The QND-ness is expressed as $Q = (P_{g,g} + P_{e,e})/2$ where $P_{g,g}$ and $P_{e,e}$ are defined as the probabilities of measuring the ground or excited state two times consecutively with the qubit initialized in the ground or excited state, respectively. To calculate the $P_{i,j}$, we apply two readout pulses, 500 ns each and separated by 1 μs (see Supplementary Fig. 5a), and then measure 1000 continuous single-shot readout traces with or without applying a previous $\pi$-pulse to the qubit (500 for each). Figure 7a shows the resulting $Q$ for the plasmon states of the IST qubit as a function of $\bar{n}$. While measurements with low $\bar{n}$ suffer from an incomplete state assignment fidelity (low SNR), in the range of measurement powers corresponding to 1000 to 1500 photons (calibrated for a steady state readout tone), we achieve the maximum QND-ness of about 92.2%. A further increase of $\bar{n}$ leads to a continuous degradation of the quantum-non-demolition character due to leakage to neighboring flux wells.

These results support the intuition that an inductive shunt can suppress leakage to non-computational states due to the absence of unbounded states above the cosine potential, as is the case for the transmon. A related effect has been shown in ref. 19 in the limit of a significantly steeper parabolic confinement. Nevertheless, more detailed simulations of the strongly driven IST-resonator system, similar to the ones presented in ref. 22, are needed to better understand and utilize the IST qubit in the high-power limit. For

example, the granular aluminum (grAl) fluxonium presented in ref. 57 is also capable of performing high-fidelity readout without the use of JPA, which was attributed to the weak nonlinearity of the grAl in comparison with Josephson junction chains. Since the geometric superinductor used in our work has exceptionally low nonlinearity[33], this might be an equally important ingredient to make the high-power QND readout work. We note that substantial further improvements in fidelity and readout time should be possible with the usual means, such as larger cavity linewidth in combination with Purcell filters.

### Preparation and decay of long-lived fluxon states

The coherence of commonly used transmon qubits—as well as the IST plasmon states—is ultimately limited by the unprotected energy relaxation time given a properly optimized design, fabrication and a well-thermalized and filtered setup. In contrast, theory predicts exponentially small matrix elements of order $10^{-13}$ for fluxon transitions in the IST qubit limit, and this is expected to result in exceptionally long fluxon energy relaxation times based on Fermi's golden rule. In the following we experimentally investigate the actual fluxon relaxation times in the IST limit.

Due to the small matrix element, we can not directly control the fluxon state, but we found that a $\approx 1$ ms long pump pulse on resonance with the readout resonator with an optimized choice of power can initialize either the ground state $|f_0\rangle$ or the first excited fluxon well $|f_1\rangle$[41,58,59], as shown in Fig. 8a. We note that this specific excitation pump power corresponding to $\bar{n} \approx 72$ for device C is significantly below the photon number used for the high-power plasmon readout in the low energy fluxon state. This indicates that there exists a

resonance condition similar to the transmon case[22] that triggers a quantum tunneling event to the higher energy fluxon well, but unlike the transmon case, this is not an ionization event due to the extended confinement potential. In the range of more than 150 photons in the cavity, the resonance condition is not satisfied (at least for the comparably short pulses used for readout), and thus the plasmon measurement can remain QND in the high readout power limit. At $\bar{n} \approx 2000$ we observe an equal probability to initialize the fluxon ground or excited state, and further increasing the resonator pump power returns the cavity to the bare cavity frequency; see Supplementary Fig. 6 for details.

Using this state preparation method, we achieve a preparation error as low as 3% for near zero external flux values in device C, as verified with an averaged plasmon frequency readout. To achieve deterministic fluxon state preparation at all flux values and independent of device parameters, we use an active state preparation scheme explained in Fig. 8b. This is followed by a fixed frequency plasmon excitation that is only successful for a certain fluxon state due to the fluxon dependent plasmon frequency. For this, we use a 3 μs long excitation pulse applied to the plasmon transition corresponding to the $|f_0\rangle$ at $\Phi_1$ (green) or the $|f_1\rangle$ at $\Phi_2$ (yellow) and $\Phi_3$ (purple) in Fig. 8c chosen to maximize the readout signal. A standard dispersive plasmon measurement in the single photon limit (to avoid measurement-induced fluxon tunneling events) is then used to resolve the specific fluxon state. This measurement procedure reveals the slow energy relaxation from $|f_1\rangle$ to $|f_0\rangle$ for three different flux bias positions, as shown in Fig. 8c.

The observed flux bias-dependent fluxon tunneling between 18 min and 3.6 h qualitatively matches with the expected dependence $\Gamma \propto \omega_{f01} e^{-\alpha\sqrt{V_{\mathrm{eff}}/E_C}} + \Gamma_{\mathrm{residual}}$ with $\omega_{f01}$ the fluxon transition frequency and $V_{\mathrm{eff}}$ representing the height of the tunnel barrier[53]. At zero flux (green color in Fig. 8c) $\omega_{f01} \approx 15\,\mathrm{GHz}$ and the potential barrier is lowered, resulting in a faster decay rate while near half flux (purple) $\omega_{f01} \rightarrow 0$ and the barrier height is nearly at its maximum of $V_{\mathrm{eff}} \approx 2E_J$ leading to much longer decay times. A quantitative understanding, in particular, with regard to the experimental limitations of the fluxon relaxation rates near half flux, such as cosmic rays and radiation, as suggested in ref. 60, or finite temperature, requires significantly more data and will be investigated in follow-up work.

## Discussion

In summary, we have theoretically and experimentally introduced a new parameter regime for superconducting qubits: the inductively shunted transmon (IST), which is characterized by very large $E_J/E_C \sim 100$ and $E_J/E_L \sim 50$ energy ratios. While the transmon is derived from the Cooper pair box circuit, the IST qubit is derived from the rf-SQUID circuit, closest to an ultra-heavy fluxonium or an ultra-high inductance flux qubit. Nevertheless, we show that the properties of the low-lying plasmon spectrum closely resemble those of the transmon but now with carefully controllable flux tunability and without charge dispersion. On a conceptual level, its potential and wavefunctions are continuous and extended in contrast to the periodic potential of the transmon. As a hallmark of this new regime, we observe stable fluxon states and quantum tunneling. The present work mainly focuses on plasmon encoding, and we identified the characteristic $E_J/E_L$ ratio as the relevant parameter to control the band dispersion and the resulting flux noise sensitivity of the device. With a demonstrated flux dispersion of only 5.1 MHz over a full flux quantum, it is significantly less noise sensitive compared to the high impedance approach investigated to date[12,13,61]. Combined with a lower flux noise amplitude inductor and lower TLS density capacitor materials, as well as an improved geometry to reduce surface loss participation, the IST concept opens a new path forward to precisely control the trade-off between flux dispersion and flux noise susceptibility[62].

In a regular transmon qubit, strong excitations, useful, e.g., for high fidelity qubit readout or stabilized bosonic qubit implementations, can easily exceed the weakly anharmonic ladder of confined states within the cosine potential. This can cause instabilities in the average number of excitations[23] and lead to excitations out of the computational basis via non-energy-conserving terms of the Jaynes-Cummings Hamiltonian[21]. The parabolic confinement of moderate inductance IST qubits based on linear geometric inductors are expected to better confine higher energy states and might be able to suppress such leakage[19]. The reported observation of quantum jumps without the need for a near quantum-limited amplifier via a high photon number qubit readout with high QND-ness and fidelity above 90%, as demonstrated at zero flux, provides supporting evidence for this hypothesis.

The fluxonium qubit platform has recently been identified as an alternative way forward to scaling up superconducting qubit processors[63,64] due to promising coherence times, higher design flexibility and anharmonicity. The use of geometric inductors could offer advantages for the reproducibility of $E_L$[13] and the current work shows that one of its major drawbacks, i.e., an enhanced flux noise amplitude, could, in principle, be mitigated with a noise-insensitive design.

Flux qubit encoding in the IST limit presents challenges due to the excessively low fluxon transition matrix elements—the reason for the observed strong protection against energy relaxation from one flux well to another. As a first step, we showed deterministic preparation of the excited or ground state and plasmon-assisted fluxon readout to monitor the energy relaxation ranging from 18 min to 3.6 h. In the future, real-time control of the qubit characteristic energies, such as the tunneling barrier $E_J$ or the effective mass $E_C$, might open a way for full phase coherent qubit control[38], as required to characterize the fluxon coherence, which we calculated to be on the order of 2 μs assuming a typical flux noise amplitude found in regular loop size SQUID devices. This could be a promising route toward new decay-protected qubit encoding schemes suitable for dynamical decoupling techniques and biased noise error correction codes.

Full control over both the plasmon and fluxon qubit encoding could lead to interesting hybrid applications in non-adiabatically driven or dynamically controlled qubit circuits that intrinsically combine fast gates with memory elements. On a more fundamental level, it might offer new capabilities to study quantum tunneling[65] in dynamically controlled potentials, and our implementation based on a ~14-mm-long SQUID wire might revive the quest for pushing the macroscopicity in superconducting quantum circuits[66–69].

## Data availability

Datasets and analysis files used in this study are available at https://doi.org/10.5281/zenodo.8004359[70].

## Code availability

Codes used in this study are available at https://doi.org/10.5281/zenodo.8004359[70].

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

## Acknowledgements

The authors thank J. Koch for discussions and support with the scQubits python package, I. Rozhansky and A. Poddubny for important insights into photon-assisted tunneling, S. Barzanjeh and G. Arnold for theory, E. Redchenko, S. Pepic, the MIBA workshop and the IST nanofabrication facility for technical contributions, as well as L. Drmic, P. Zielinski and R. Sett for software development. We acknowledge the prompt support of Quantum Machines to implement active state preparation with their OPX+. This work was supported by a NOMIS foundation research grant (J.F.), the Austrian Science Fund (FWF) through BeyondC F7105 (J.F.) and IST Austria.

## Author contributions

F.H. developed the concept and theory, designed and fabricated the devices, performed the measurements and analyzed the data. M.P. contributed to the design, fabrication and data analysis. L.N.K. contributed to measurement and data analysis. A.T. contributed to the design and fabrication of the devices as well as the theory and concept. M.Z. contributed to the design and fabrication. J.M.F. supervised the project.

## Competing interests

The authors declare no competing interests.
