## [Peer Review File · Nature Communications]

The manuscript titled “A superconducting qubit with noise-insensitive plasmon levels and decay-protected fluxon states” attempts to define a new qubit and propose to use it in place of the tunable transmon. The authors provide a few examples when this platform can be useful, and follow suit with simulation details and experimental data. I believe that the temperature-dependence of fluxon tunneling can be useful in exploring phase slip physics in the future. The theory description is also well written. However, the novelty of this platform is questionable, and I believe it is not suitable for publication in a journal with broad readership such as Nature Communications.

Let me simply discuss the points that I found to be unsubstantiated.

1/ The authors claim that the plasmon levels are noise-insensitive. However, their coherence times are not good. As far as I understand, these levels are similar to transmon in terms of T_1 protection, and since T_2 cannot be higher than $2T_1$, these should not be considered “protected”.

2/ The authors claim that the plasmon levels can be used instead of flux-tunable transmons. However, I don't see the point of having the flux dispersion equal to zero either. Although the qubit is insensitive to flux noise in this regime, its frequency cannot be tuned by flux either. Hence, its usage is limited.

3/ The authors claim that the IST's anharmonicity is better than transmon's. Yet, in plot 2(c) and (d), it is clear that in the proposed regime, the anharmonicity is comparable to transmon's.

4/ The authors claim that excitation out of the cosine potential limits the readout power in transmon system. This is somewhat misleading. As far as I understand, the QNDness will decrease the readout fidelity substantially (at the level of a few photons, [T. Walter et al., Phys. Rev. Applied **7**, 054020]) before excitation outside of the cosine potential kicks in (at the level of ~ 100 photons, [R. Lescane et al, Phys. Rev. Applied **11**, 014030]). The manuscript cites the paper by D. Sank et al., which explains non-QND effect with induced state transitions, which hardly relates to how the inductive shunt would solve the problem. Recently, [R. Shillito et al., <https://arxiv.org/abs/2203.11235>] made a step forward in identifying the cause for non-QNDness at the few photons level, while the cosine potential is discussed in [R. Lescane et al, Phys. Rev. Applied **11**, 014030].

5/ The authors claim that the qubit can be useful for cat experiments. However, they probably cite the wrong paper. The correct references should be for dissipative cats. The Kerr cat is a pumped non-linear oscillator requiring a specific ratio between third and fourth-order coefficients. It does not need an ancilla at all, and there is no transmon involved in that experiment (Ref. 30).

6/ Decay-protected states with no coherence (or rather, no indication of coherence), such as the fluxons in the proposed regime here, behave similarly to classical bits.

7/ The measured spectrum clearly shows flux dependence, and it is not much different from that in heavy-fluxonium papers (Y. Lin et al., Phys. Rev. Lett. 120, 150503 and N. Earnest et al., Phys. Rev. Lett. **120**, 150504 both appeared on arXiv in 2017 then published in 2018)

8/ The authors do not provide rigorous comparison between the IST and, e.g., C-shunt flux qubits, the quarton, or the SNAIL. It will be unclear to the readers what makes the proposed platforms superior to these in terms of modification of the cosine potential with the inductance. For example, the quarton paper [<https://arxiv.org/abs/2006.04130>] discusses modification of the potential by tuning the three energy scales.

9/ The experimental methods are not substantially different from previous works by the authors. I understand that geometric superinductors are interesting and impressive, but the authors have already published quite nice papers on this subject.

I encourage further works to substantiate the points that motivate exploration of this qubit. Before that, I cannot recommend this to appear on Nature Communications.

Reviewer #2 (Remarks to the Author):

Summary

The manuscript presents a new type of superconducting qubit, the inductively shunted transmon (IST), formed by shunting a transmon qubit with a large superinductor made using a long coil of Aluminum wire. The parameters are chosen to investigate a previously unexplored regime of $E_J/E_C \gg 1$ and $E_J/E_L \gg 1$. This design makes the plasmon levels completely insensitive to charge noise while reducing flux dispersion as well. The authors present a detailed theoretical description of IST along with its relationship to fluxonium and transmon. The experiments are performed on three chips made using Aluminum circuits on Silicon wafers. The coherence properties at the base temperature are moderately good and mostly stable across different flux-bias points and with time. Two of the devices have flux dispersion of only 5 MHz while another one has about 40 MHz due to larger inductive energy. The authors perform flux sweeps at different temperatures to show the tunneling events activated by thermal energy. Next, relaxation and dephasing times are studied along with a model to explain those values. The paper ends with discussions on a few strategies to improve the performance of IST qubits and future studies possible with IST-like devices.

Review

The paper is extremely well written, and the results are presented in a clear and intuitive fashion with supporting figures. The elegance of this work comes from the successful realization of the large superinductor that enabled reaching a high Josephson energy to inductive energy ratio. These new parameters resulted in complete insensitivity to charge noise and suppressed flux-noise sensitivity of the IST. While I believe the science presented here is sound and interesting, IST is not going to be adopted as a mainstream qubit in near future. I would like the following comments to be addressed before the manuscript can be accepted for publication.

-Title: The manuscript primarily deals with the plasmon states. While the flux-sweep data indirectly indicates the protection of the fluxon states against relaxation, no data is presented to demonstrate the use of the fluxon states as qubits. In fact, the preparation of the fluxon states would be very challenging for this device. Hence the title should focus on the plasmon levels only.

-Abstract: “sweet spot everywhere for a flux tunable device” – this sentence is slightly misleading because typical tunable transmons can be tuned by at least several hundreds of MHz and flux-insensitivity of IST comes from the very fact of reduced flux dispersion (about 5 MHz). I believe saying “mildly flux tunable device” would be more appropriate.

-Fig. 2a: Add 0 to y-axis so that readers have a clear idea of how small the fluxon transition is.

-Fig. 2, caption: Change “flux and plasmon transition” to “fluxon and plasmon transition”

-Page 2, left paragraph: The authors argue that charge dispersion limits the use of transmon as a qubit. But the same could be true for IST as the higher levels of IST could be exponentially more sensitive to flux noise. Ideally, the authors should measure the flux-dispersion for the second transition or at least present theoretically expected behavior.

-Fig. 3, caption: I would modify “Higher inductance leads to smaller phase confinement...” to “Higher inductance leads to weaker phase confinement..”

-Page 4, top-left paragraph: Change “flat vs. flux” to “flat with respect to flux”.

-Page 4, Eq. (2): The second-order derivative of ω_{p01} with respect to flux was not shown. Maybe add the expression to Methods or to supplement.

-Page 4, Table I: Showing Hahn echo decay constants would be useful to understand the spectral content of the flux-noise.

-Page 4, bottom-right paragraph: Change κ to $\kappa/2\pi$ for accuracy.

-Fig. 5, caption: Change the order either in the sentence “show a fit to the plasmon qubit and readout resonator” or the figures (c) and (d).

-Fig. 5b: Why does the x-axis start from 300 mV here? Is the current used as a heat source?

-Page 5, right column: It was not clear what the authors meant by “This is extreme case ... control schemes”. A little more elaboration would be useful.

-Fig. 6, caption: Given the base temperature of 7 mK and a strong effort to thermalize, it is unusual to have a bath temperature of 108 mK. Some discussion on how the authors arrived at this number would be useful.

-Page 6, bottom-left paragraph: “Combining a set of independent flux sweep...” – How are the independent sweeps determined, different starting points in flux or temperature?

-Page 6, bottom-right paragraph: Refer to figure 3 while mentioning $|p1\rangle$ as it is not defined earlier.

-Page 7, left column: How is the flux-noise amplitude calculated to be $98 \mu\Phi_0$?

-Page 9, after Eq. (13): Typo, the energy difference between first and second excited should have $+2E_C$ in the expression.

-Page 9, Eq. (15): Change the equality to approximately.

-Page 9, Device fabrication: The length of the geometric superinductor would be an important number to mention. Since the performance of the device depends very strongly on the fabrication, it would be great if the key steps are shown with a figure along with important precautions taken. Was there a particular reason to use the air-bridges, as one could modify the geometry to a simple meander and avoid additional fabrication steps?

-A full circuit diagram including different attenuators and filters used in the experiment would be beneficial for the readers.

Additional comments:

-Fig 3b: The colors of traces corresponding to $EL = 2$ and 0.25 look very similar. Changing the colors to improve contrast would be great.

-Fig 3d: The discontinuities in the inset look odd.

-Page 4, Table I: Since the authors made a comparison with transmons several times, it would be nice to have all the coherence and dispersion properties for the ω_{p12} transition as well.

Reviewer #3 (Remarks to the Author):

The article "A superconducting qubit with noise-insensitive plasmon levels and decay-protected fluxon states" by F. Hassani, et al. describes the theory and implementation of an inductively shunted transmon. The inductively shunted transmon regime is defined as a special type of rf-SQUID for which E_J (Josephson energy) $\gg EC$ (charging energy) and $E_J \gg EL$ (inductive energy). It can be seen as the first transition of a heavy fluxonium (Earnest PRL 2018) at its upper sweet spot with a very large Josephson energy or as a transmon with a large inductive shunt. In this regime, the first plasmon transition resonates near the plasma frequency $\sqrt{8 EJ EC}$ regardless of the applied external magnetic flux. This property results in a protection against flux noise in addition to the built-in charge noise protection provided by the large inductance shunting the Josephson junction.

The idea of an inductively shunted transmon is not new. As it is stated in the introduction of the paper, it was previously proposed, for instance, to create a new readout coupling scheme (Richer PRB 2017) or to prevent instability with large off-resonant pump amplitude (Verney PRApplied 2019). However, as far as I am aware, the parameter regime studied here has no experimental counterpart in the literature. The topic is relevant and timely as there are significant efforts in the superconducting circuit community aimed at creating new kind of scalable circuits with potential to outperform the widespread transmon circuit (see e.g. Nguyen PRL 2018, Pechenezhskiy Nature 2020, Gyenis PRX Quantum 2021, etc) or to use alternative circuit elements to improve existing circuits (see e.g. Grunhaupt Nature Materials 2019, Peruzzo PRApplied 2020, Place Nature Communications 2021, etc).

The paper is overall well written and sounds very accurate scientifically. The fabrication of the device is impressive and quite unique. All the theory and experimental details are clearly provided in the main text or in the methods. The new qubit regime realized here will certainly trigger a lot of excitement in the community as it might hold the solution to several technological challenges faced

by the community. However, there are several issues and questions that need to be fully addressed before one can consider the paper for publication in a journal like Nature Communications.

Science Questions

Could the authors comment on the stability of the circuit ? The authors observe fluxon tunneling on time scales of the order of minutes. This is surely quite impressive and it demonstrates once more the validity of Fermi's golden rule for the rf SQUID. However, thermal excitations of the diamond shape transitions seem unavoidable close to $\phi_{ext} = \pi$. This is both due to the low frequency of the transition and to the fact that the phase matrix element of this transition becomes maximal at this particular flux point. Will such excitations compromise the stability of the circuit used as a qubit ? How is the frequency of the IST transition affected by the tunneling of a fluxon ? I think it would be good to clarify the role of these transitions when considering the circuit to store and manipulate quantum information.

The authors say that 'high fidelity excited state preparation is possible' when talking about fluxons. What does high-fidelity mean here ? How is the fidelity estimated ? How is the excited state prepared ? I think this sentence needs to be clarified.

Is there a limitation on the minimal frequency of the IST for a given inductance ? It seems that to fulfill the IST condition, the qubit frequency has to be relatively large (typically 5 GHz or more) or that the inductance has to be increased significantly. The authors should maybe comment on the order of magnitude of frequency and coupling strength that can be achieved with such a circuit.

Readability and message

Overall, I think that the impact of the paper would be greatly improved if it was more targeted towards the new results presented by the authors. The paper starts with a general review of the rf SQUID properties. In particular, the comparison between the different regime (inductively shunted transmon, transmon, fluxonium, Blochonium) takes the first three figures with very long captions as well as the first three pages of the paper. The actual device – which is the topic of the paper – is only introduced in Figure 4. While I understand the motivation of the authors, I would suggest presenting the IST directly without explaining again the spectrum of the fluxonium and the transmon (Fig2 a,b,e Fig 3 a lower panel), which is usually assumed to be known in transmon/fluxonium papers.

The authors should avoid using jargon such as "circuit's desire", "plasmonic character", "we solve the Hamiltonian", "dissipative leakage", "decompactifies the phase". I think that removing such expressions would avoid any ambiguity.

The authors make some claims which are not well supported by experiments yet and which dilute the message of the paper without adding much to the content. See my comments below.

The authors claim that their approach “offers a new path forward towards improved qubit encodings in single and multi-mode circuits” in the abstract. I don’t see how the approach is new. The parameter regime of the circuit is new but the approach consisting in controlling matrix elements and sensitivity to noise sources with circuit parameters and design is certainly not new (see e.g. Lin PRL 2018, Earnest PRL 2018).

The authors claim that the circuit can be used as a qudit as higher levels are also protected from charge noise. But does the flux noise protection decrease with level number ? I find this claim very hand-wavy since the lifetime and coherence time data of higher levels of the circuit are not presented in the paper.

The authors claim that their circuit will be compatible with large drive amplitude for high-power readout and parametric stabilization of bosonic codes. Indeed, some studies anticipate that circuits with a bounded potential are more robust to large drive amplitudes. However, as far as I am aware, this claim is not yet supported by experimental evidence. There are both experimental observation showing a very poor Quantum Non-Demolition character of the readout of fluxonium (see e.g. Ficheux PRX 2021) and relatively good performances (see Gusenkova PRApplied 2021). There is even no clear numerical evidence that this is the case. The paper cited by the author (Verney PRB 2019) suggests that it is the case but it does not access the transient dynamics of the field ramping up in the cavity, which is crucial to understand the process (see arXiv:2203.11235). The claim is also beyond the scope of the paper.

Other issues

The authors use the terminology ‘sweet-spot everywhere’ in the abstract. I find it disturbing since a sweet spot has a well established mathematical definition (see Gyenis PRX Quantum 2021) i.e. that the first order derivative of the energy with respect to a given parameter exactly cancels. I understand what the authors meant: the flux sensitivity is strongly suppressed by the choice of parameters as given by Eq. (2). But I would avoid calling it a sweet spot away from integer and half integer flux points.

The authors say that their circuit exhibits 'ultra-small' matrix elements (caption Fig 2). Small compared to what ? Later on in the main text, it said that the matrix element is in the range $10^{(-13)}$. Are we talking about the charge matrix element as represented in Fig 2 d ? Is that value extracted from numerical diagonalization ? Isn't it limited by numerical errors ?

Minor issues

I think that the acronym IST is first used without being defined.

The x-label of Fig 3 d is shifted to the right.

In Fig 3 a, why are the odd level wave functions mirrored with respect to $\phi=0$ between the transmon and IST ?

The inset of Fig 3 d is discontinuous, is it because of the avoided crossing with the fluxon lines ?

Typo: Methods, Fitting procedure, "einenenergies"

Typo: p7 left column last paragraph "it's"

Caption Fig4 missing space after "d."

Reply to all referees

We thank all three reviewers for taking the time and for giving us valuable feedback to improve our work and its impact on the community. We identified two major points common to the reports, which we address with new experimental data that is presented in two new subsections: “*High-power QND qubit readout*” as well as “*Preparation and decay of long-lived fluxon states*”. To accommodate the new results, we changed the title, abstract, introduction and the conclusion somewhat. All changes and additions to the main text are highlighted with red color.

In the following, we respond to all reviewer comments in detail. Answers are in blue and related changes to the manuscript are in red font.

Reviewer 1:

The manuscript titled “A superconducting qubit with noise-insensitive plasmon levels and decay-protected fluxon states” attempts to define a new qubit and propose to use it in place of the tunable transmon. The authors provide a few examples when this platform can be useful, and follow suit with simulation details and experimental data. I believe that the temperature-dependence of fluxon tunneling can be useful in exploring phase slip physics in the future. The theory description is also well written. However, the novelty of this platform is questionable, and I believe it is not suitable for publication in a journal with broad readership such as Nature Communications.

We thank the reviewer for the time and evaluation of our work. We hope that our response below as well as the new data will convince him/her that we are not only realizing a new regime of superconducting circuits but also novel physics.

We are quite certain that the large part of the field that attempts to scale-up transmon devices will eventually hit a roadblock when it comes to applicability and usefulness due to dramatic error correction overheads and control complexity. New concepts need to be developed and shared with a large audience to trigger the crucial ideas required to keep the momentum and impact of superconducting qubits going. We consider our work a significant contribution to this effort by exploring new methods (linear superinductors) and new parameter regimes (see Fig. 1) with surprising results, such as no charge noise of higher levels, by far the lowest flux dispersion in any flux tunable qubit, >3 hours of fluxon decay times, and QND single shot readout without a Josephson amplifier, to name a few.

1) The authors claim that the plasmon levels are noise-insensitive. However, their coherence times are not good. As far as I understand, these levels are similar to transmon in terms of T_1 protection, and since T_2 cannot be higher than $2T_1$, these should not be considered “protected”.

We want to start with an example why coherence times are not the most relevant quantity to look at to judge the potential of a new qubit concept or specific method. When the fluxonium was invented higher coherence was not one of its features, in fact there was no coherence time data

at all in the seminal Science paper. Now, standard fluxonium qubits are also considered for larger scale devices due to important advantages related to anharmonicity and coherence.

Having said that, we will now outline what protection means, we make quantitative comparisons with the state of the art and explain why the noise insensitivity we achieved is unique.

Intrinsic protection in superconducting qubits is divided into two fronts with one being the protection from relaxation and the other protection against dephasing due to noise. We agree that the discussed plasmon states of the qubit are not protected against dielectric losses and relaxation and the main text specifically states this for plasmon levels: "The relatively high matrix element and transition frequency of the plasmon state render it susceptible to dielectric losses and the observed variation indicates possible two-level-system coupling."

However, we want to emphasize that the focus of this work (in case of the plasmon encoding) is on the second front with protection against noise sources. And here it is important to understand that the protection of a qubit with respect to a noise source is not judged by the value of T2 but the dependence of the T2 values with respect to the magnitude of that noise source (on top of that the T2 value itself can still be limited by other means such as 2T1 or photon shot noise as is stated in the main text and by the reviewer). Given the 100 times larger flux noise amplitude (due to the longer geometric inductor wire), achieving a T2 that is not limited by flux noise away from the sweet spot for the first time in any SQUID device is a surprising, significant and important result for our community.

For any flux tunable device the intrinsic and therefore most relevant source of noise - is flux noise due to magnetic dipoles on the surface. Fig 6.c proves the claim of noise insensitivity with a T2 of about 13 μ s over an entire flux quantum, which is unique among any reported flux tunable devices reported in the literature.

To support this statement, we provide the T2 dependence of the state of the art (highly optimized) loop-type qubits for comparison - all of them show strongly reduced T2 even slightly off the half flux point, in some cases even when an echo sequence is used:

- 1) Nguyen, Long B., et al. "High-coherence fluxonium qubit." *Physical Review X* 9.4 (2019): 041041. (Fig 4.a)

- 2) Gyenis, András, et al. "Experimental realization of a protected superconducting circuit derived from the 0- π qubit." *PRX Quantum* 2.1 (2021): 010339.(T2R in Fig.6)

- 3) Zhang, Helin, et al. "Universal fast-flux control of a coherent, low-frequency qubit." *Physical Review X* 11.1 (2021): 011010. (Fig 3.b)

Another measure for flux noise sensitivity is the flux dispersion, which refers to the range of qubit frequencies over a full flux quanta range and here is another comparison with the literature:

- Capacitively shunted flux qubit: a few GHz
- Fluxonium: a few GHz
- Soft zero_pi: a few GHz
- Quasi charge qubit (Blochonium): 100 MHz
- IST qubit: 5 MHz

We furthermore show in the inset of Fig. 6a that the T1 equivalent quality factor (representing the losses in the inductor and capacitor pads) is of the order of 670,000 which matches well with state of the art fabrication of aluminum devices on silicon at 6 GHz. In this context it is important to realize that many of the high T1 qubits operate at much lower frequencies where the dielectric loss is correspondingly smaller.

Having said that, of course there is also room for improvement by optimizing the geometric design to reduce surface participation (no such optimization has been done for this device), and by applying new materials such as tantalum (Place, Alexander PM, et al. *Nature communications* 12.1 (2021)), changing or engineering the substrate such as using sapphire (Somoroff, Aaron, et al. *arXiv:2103.08578*) or using (large) vacuum gaps (Zemlicka, M., et al. *arXiv:2206.14104*), and finally to shield and thermalize the input lines to achieve lower photon shot noise.

Most examples above are technical and in some sense incremental improvements that apply to any qubit experiment. Our work instead shows a conceptual advance in a first generation, proof of principle device. We show that the low dispersion plasmon transitions of the IST qubit can fully suppress flux noise in our experiment and we achieve that with low characteristic impedances that are relatively easy to fabricate (in contrast to the Blochonium that comes closest - but does not show flux dependent coherence).

In summary, the reported coherence times are very encouraging, relatively well understood and we are convinced that in the future the IST qubit can not only compete but overcome that of flux tunable transmons with the same engineering improvements. The IST plasmon levels have no charge offset sensitivity and are flux noise insensitive in the same sense as the transmon levels are charge noise insensitive (even though not exponential). In our opinion this is communicated clearly in the manuscript now - in particular in the title - and we have made further improvements to the manuscript to avoid any confusion in this regard:

Title: We now state that the plasmon qubit is *flux-noise insensitive* rather than *noise insensitive* since we expect that photon shot noise still affects it similar to the transmon.

We added the references listed above to the sentence: “Given the two orders of magnitude higher flux noise amplitude the result shown in Fig. 6c represents a new level of dephasing protection in comparison with the most coherent flux tunable qubits [39, 51, 63] away from the flux sweet-spot.”

2) The authors claim that the plasmon levels can be used instead of flux-tunable transmons. However, I don't see the point of having the flux dispersion equal to zero either. Although the qubit is insensitive to flux noise in this regime, its frequency cannot be tuned by flux either. Hence, its usage is limited.

Current quantum processors have very strict limitations for the frequency range and operating position of each qubit to avoid frequency crowding and small tuning ranges are of very high value in this context, see for example Chávez-García, José M., et al. "Weakly Flux-Tunable Superconducting Qubit." *arXiv:2203.04164* (2022) for a recent proposal.

This is true not only (but in particular) in the case of cross-resonance gate architectures where laser annealing has been developed to perform the fine tuning of the individual Josephson junctions after fabrication and as far as we know this is a crucial step in enabling IBM to develop larger scale devices as otherwise the probability to find a workable lattice without frequency collisions quickly approaches zero. An in-situ tuning knob with a tuning range matching the typically best junction fabrication variation of around 5-10% across a chip is thus an ideal situation that our approach can provide.

The conventional way of making a tunable transmon is to have a split junction with symmetric or asymmetric Josephson energy (to define 2 sweet spots and limit the flux dispersion). Due to the nature of Josephson junction fabrication and its tunnel barrier, designing for a specific dispersion is challenging however. In the IST qubit the dispersion is controlled by the inductive shunt and through Eq. 2 so any useful dispersion with external flux can be achieved controllably simply by changing the geometry (e.g. the number of turns) of the coil in case that inductor type is used. The IST qubit allows for careful engineering of the trade-off between flux tunability and flux noise insensitivity.

We clarify this point by changing this sentence in the second paragraph of the Discussion section: "...the IST concept opens a new path forward to introduce in-situ fine tuning of the transmon frequency without sacrificing protection against flux noise." to "...the IST concept opens a new path forward to precisely control the trade-off between flux tunability and flux noise insensitivity [70]."

3) The authors claim that the IST's anharmonicity is better than transmon's. Yet, in plot 2(c) and (d), it is clear that in the proposed regime, the anharmonicity is comparable to transmon's.

We do not claim that IST qubit anharmonicity is better than that of the transmon. Fig. 2c clearly shows that the anharmonicity of the IST qubit at best converges to that of the transmon in the high E_J/E_I limit. As the inductive shunt decreases, the quadratic term of the potential in Eq. 1 becomes dominant over the cosine term and therefore the circuit acts as harmonic oscillator, which is the reason for the reduction of anharmonicity. This is explained in the main text on page 4, the paragraph before the experimental realization.

4) The authors claim that excitation out of the cosine potential limits the readout power in transmon system. This is somewhat misleading. As far as I understand, the QNDness will decrease the readout fidelity substantially (at the level of a few photons, [T. Walter et al., Phys. Rev. Applied 7, 054020]) before excitation outside of the cosine potential kicks in (at the level of ~ 100 photons, [R. Lescane et al, Phys. Rev. Applied 11, 014030]). The manuscript cites the paper by D. Sank et al., which explains non-QND effect with induced state transitions, which hardly relates to how the inductive shunt would solve the problem. Recently, [R. Shillito et al., <https://arxiv.org/abs/2203.11235>] made a step forward in identifying the cause for non-QNDness at the few photons level, while the cosine potential is discussed in [R. Lescane et al, Phys. Rev. Applied 11, 014030].

To our understanding, the references “R. Shillito et al.” and “T. Walter et al.” and “D. Sank et al.” agree that the decrease in QND-ness as a function of measurement power has roots in terms ignored by the rotating wave approximation of the Jaynes-Cummings Hamiltonian, and “R. Shillito et al.” in addition shows that depending on the parameters the ionization can occur with the resonator occupied by only a few photons leading to a non QND measurement and explaining the observations of “T. Walter et al.”. Moreover, in “R. Shillito et al.” they confirm that their simulation does agree well with steady state simulations performed by “R. Lescane et al.”, meaning that what is called “structural instability” and “ionization” are basically the same. Following the work of “R. Lescane et al.”, “Verney, Lucas, et al. *Physical Review Applied* 11.2 (2019): 024003.” shows that the inductive shunt can avoid the ionization and extend the limit for high power measurement.

Our new measurement data shown in the new Fig. 8 also confirms that in case of the device with the highest confinement (highest EL, lowest inductance) we can increase the photon number to about 600 photons in the cavity to perform single shot qubit measurements without the use of a parametric amplifier with high QND-ness and fidelity. Similar to the case of the coherence times, this quite remarkable result has been achieved without any optimizations with regards to the dispersive shifts and cavity readout linewidth and without the use of Purcell filters. We are not aware of this in any transmon or fluxonium device based on junction arrays.

We note that the linearity and single phase character of the geometric inductor are very likely also needed to achieve this fairly remarkable result and further experimental and numerical studies into the power handling as a function of confinement potential would be useful and interesting but go beyond the scope of this work. The latter will also be needed to quantitatively explain the employed fluxon state preparation schemes in the second new manuscript subsection.

We added the new data of high fidelity QND readout in a new section denoted “High-power QND qubit readout”. The references for “R. Shillito et al.” and “Verney, Lucas, et al. *Physical Review Applied* 11.2 (2019): 024003.” were added accordingly.

5) The authors claim that the qubit can be useful for cat experiments. However, they probably cite the wrong paper. The correct references should be for dissipative cats. The Kerr cat is a pumped

non-linear oscillator requiring a specific ratio between third and fourth-order coefficients. It does not need an ancilla at all, and there is no transmon involved in that experiment (Ref. 30).

The referee is correct - our work applies only directly to the dissipative implementations typically based on a transmon ancilla. We have corrected the reference and now cite the recent review by Ma, et al., Science Bulletin, 2021 <https://doi.org/10.1016/j.scib.2021.05.024>.

6) Decay-protected states with no coherence (or rather, no indication of coherence), such as the fluxons in the proposed regime here, behave similarly to classical bits.

Clearly this qubit circuit is not in a classical regime. Fluxons are a valid and stable qubit encoding in this circuit that is initialized in its quantum ground state and not driven. While the T2 sensitivity of the fluxon state is higher than in a usual heavy fluxonium, it is far from any classical limit as we show below. In fact, having to deal with no bit flip errors can hugely simplify error correction in larger processors even if the dephasing occurs faster, see e.g. <https://doi.org/10.1103/PhysRevLett.120.050505>.

As a general comment, a large wavefunction overlap is not required to observe quantum systems in superposition states - think of an EPR pair. It does however help to prepare them in such a superposition state and we were very careful in our manuscript not to misrepresent this fact. Also, we provide numbers and methods below that encourage the use of these long lived states to encode quantum information in the near future. In our opinion there is not only really interesting physics here but there are also interesting opportunities that might have a big impact that goes beyond incremental improvements of existing encodings.

In this revision, we included a new section named “Preparation and decay of long-lived fluxon states” that includes initialization and direct time-dependent measurements of the fluxon level energy relaxation for three flux bias positions. We observe relaxation times ranging from 18 minutes to 3.6 hours.

These measurements are novel not only due to the long observed lifetimes but also because of the new methods i.e. a very efficient and fast fluxon readout via the well-dependent plasmon frequency spacing that affects the superconducting resonator via the dispersive interaction. This is a new paradigm compared to phase qubit experiments that were conducted with a direct phase slip readout based on tilting the potential by directly applying a destructive direct current to the device.

The observed decay-protected fluxon states in the IST limit have several important features that makes them experimentally interesting for the community:

- 1) The long relaxation is pretty much unprecedented and it is an interesting question what the fundamental limitations of the quantum tunneling dynamics are. Such tunneling experiments and its quantitative modeling goes beyond the scope of this work but we plan to study this systematically in the near future.

- 2) These long energy relaxation times are achieved in a non-driven system, meaning that the longer T_1 is not directly related to a lower T_2 which to the best of our knowledge is the case in driven-dissipative bosonic qubit encodings.
- 3) In the present work we don't show coherent control of the fluxon state, however this does not mean that coherence for the IST fluxon state fundamentally does not exist or is experimentally impossible to control. Considering the derivative of the fluxon state as $\delta\omega_{f01}/\delta\Phi_e \simeq 2\pi EI$ and using a typical flux noise amplitude of $1 \mu\Phi_0$ in Josephson junction chain RF SQUIDs one can predict a flux noise limited T_2 of $2 \mu\text{s}$ for the fluxon state. One can see that the IST benefits here from a very small EI (high inductance) and it is very conceivable to lower it further.
- 4) In the future we plan to use a variation of the IST qubit with the possibility to either dynamically control the potential barrier height (tunable E_j) or control the mass of the particle (controlling E_c), as shown in the circuit diagrams below, to prepare a fluxon state superposition.

dynamically controlling the potential barrier dynamically controlling E_c

- 5) Since there are several methods to deal with flux noise (which is intrinsically of lower magnitude compared to charge noise) and due to its beneficial $1/f$ dependence that allows for applying CPMG pulses and refocusing, having a qubit where the T_1 limit is orders of magnitude longer is a very valuable starting point for further improvements.
- 6) The longest decay rate reported so far in superconducting qubits has been obtained in a heavy fluxonium (8 ms reported in Earnest et al., Phys. Rev. Lett. 120, 150504), benefiting from the double well potential configuration and suppressed matrix elements of the fluxon transition. At the moment it is not clear how much further superconducting qubits can continue in improving T_1 by exploiting Fermi's golden rule. Questions such as coherent control or whether there are other limiting factors for T_1 (such as cosmic rays, quasi-particles, finite temperature) remain open. The IST qubit allows the study of these open questions in the future.

7) The measured spectrum clearly shows flux dependence, and it is not much different from that in heavy-fluxonium papers (Y. Lin et al., Phys. Rev. Lett. 120, 150503 and N. Earnest et al., Phys. Rev. Lett. 120, 150504 both appeared on arXiv in 2017 then published in 2018)

In both cited papers the spectrum is periodic which is a fundamental difference to the IST qubit. In the Hamiltonian of a loop device (Eq.1), the cosine term modifies the potential periodically as Φ external changes. Therefore until now it was expected that the spectrum of a loop device to be periodic with respect to Φ external. Our work shows a non-periodic spectrum for a loop

device, explains the discontinuities based on quantum tunneling, shows a transition between non-periodic spectrum to a periodic one by controllably increasing the tunneling rate, retrieves the flux quanta and explains a fitting procedure for such a spectrum. All these aspects are different and novel. Additionally, both papers cited by the reviewer are focused on fluxon transitions and their coherence properties whereas in this paper we study mainly the properties of plasmon transitions and their intrinsic protection against flux noise. Also, the limit of the cited papers are far from the IST limit, for example the paper N. Earnest et al., Phys. Rev. Lett. 120, 150504 is indicated as heavy fluxonium in Fig.1 and referenced. We emphasize that the fluxon transition matrix element in our work is 9 orders of magnitude smaller.

8) The authors do not provide rigorous comparison between the IST and, e.g., C-shunt flux qubits, the quarton, or the SNAIL. It will be unclear to the readers what makes the proposed platforms superior to these in terms of modification of the cosine potential with the inductance. For example, the quarton paper [<https://arxiv.org/abs/2006.04130>] discusses modification of the potential by tuning the three energy scales.

In our paper we do discuss the difference between the flux qubit (with a small inductive shunt where inductive energy is comparable with Josephson energy), i.e. its potential, wavefunction and dispersion in comparison with IST qubit (with a large inductive shunt or high EJ/EL). **We have added the original flux qubit to Fig. 1 and kept the c-shunted flux qubit now correctly named. We also added the sentence: "We note that the quarton qubit and the SNAIL element are quite close to the flux and c-flux qubit respectively in this parametrization [37, 38]." in the caption to clarify where those circuits fit it without crowding the figure. They both operate in the low EJ/EL, high EJ/EC limit similar to the flux qubits.**

9) The experimental methods are not substantially different from previous works by the authors. I understand that geometric superinductors are interesting and impressive, but the authors have already published quite nice papers on this subject.

Here, the geometric superinductors were used as a versatile tool to explore new parameter regimes and unfold new techniques to suppress flux noise. Once a useful superinductor technology is developed it is expected to be used in different applications. The reviewer's comment is also valid for Josephson chains and other superinductors. Since 2007, when the first Josephson array paper was published, the experimental techniques of realizing the chains are pretty much the same. That does not decrease the value of the works using the Josephson arrays, but is a testament to the reliability of the chains as a tool to explore superconducting qubits.

Having said that, the linearity and the absence of phase slips or critical current noise is an interesting aspect that might contribute to the exceptional fluxon stability and power handling for readout that we are observing in these devices. We have not seen similar readout capabilities or fluxon lifetimes in any JJ-chain qubit. So here the method is still a matter of new insights and therefore a plus in terms of novelty.

I encourage further works to substantiate the points that motivate exploration of this qubit. Before that, I cannot recommend this to appear on Nature Communications.

We hope that our reply and the substantial amount of new material that we have added to the paper will convince the referee that the IST qubit represents an interesting platform not only for further exploration but also for other readers of Nature Communications.

Reviewer 2:

The paper is extremely well written, and the results are presented in a clear and intuitive fashion with supporting figures. The elegance of this work comes from the successful realization of the large superinductor that enabled reaching a high Josephson energy to inductive energy ratio. These new parameters resulted in complete insensitivity to charge noise and suppressed flux-noise sensitivity of the IST. While I believe the science presented here is sound and interesting, IST is not going to be adopted as a mainstream qubit in near future. I would like the following comments to be addressed before the manuscript can be accepted for publication.

We thank the referee for these encouraging words and the very detailed suggestions and corrections. We did our best to fully address all comments below.

1) Title: The manuscript primarily deals with the plasmon states. While the flux-sweep data indirectly indicates the protection of the fluxon states against relaxation, no data is presented to demonstrate the use of the fluxon states as qubits. In fact, the preparation of the fluxon states would be very challenging for this device. Hence the title should focus on the plasmon levels only.

We added a new section called “Preparation and decay of long-lived fluxon states” that includes both a deterministic initialization to one of its basis states and time-dependent energy relaxation data. Such direct single-shot measurements of the fluxon decay demonstrate an average of 3.6 hours of energy relaxation time for a flux bias that is close to half-flux. With such long time scales we had to measure for 2 weeks to get sufficient statistics.

We believe that this new data justifies the title. We in fact added the directly observed energy relaxation time in this revision to make it more specific. We have changed the title to “Inductively shunted transmon: A superconducting qubit with flux noise insensitive plasmon states and a protected fluxon decay exceeding 3 hours”. We also made it clear in the abstract that we do not yet achieve full phase coherent qubit control in the fluxon basis and in the conclusion make it a bit more concrete how we plan to get there (see also answer 6 to reviewer 1).

2) Abstract: “sweet spot everywhere for a flux tunable device” – this sentence is slightly misleading because typical tunable transmons can be tuned by at least several hundreds of MHz and flux-

insensitivity of IST comes from the very fact of reduced flux dispersion (about 5 MHz). I believe saying “mildly flux tunable device” would be more appropriate.

This term was introduced and used by the inventors of the transmon and refers to the fact that the qubit levels vs. charge become so flat as a function of E_j/E_c that the charge sweet spot stretches over the entire range of charge bias values, i.e. the Cooper pair box becomes useable at any charge value. We note that such a qubit (the transmon) is also NOT tunable in charge. In the same sense, in our case the plasmon bands vs. external flux become flat enough to use the (flux-type) qubit at any external flux value (for comparison to other work see our answer 1 to referee 1).

We would go with the suggestion of the referee but, the term seems to cause confusion as it is also criticized by referee 3 so we removed it all together. Instead we now write: “In this work we introduce the inductively shunted transmon, a weakly flux tunable superconducting qubit that offers charge offset protection for all levels and a 20-fold reduction in flux dispersion compared to the state-of-the-art resulting in a constant coherence over a full flux quantum.” which we hope is more to the point.

3) Fig. 2a: Add 0 to y-axis so that readers have a clear idea of how small the fluxon transition is.

We added 0 to the y-axis of Fig.2a.

4) Fig. 2, caption: Change “flux and plasmon transition” to “fluxon and plasmon transition”

This has been corrected as suggested.

5) Page 2, left paragraph: The authors argue that charge dispersion limits the use of transmon as a qubit. But the same could be true for IST as the higher levels of IST could be exponentially more sensitive to flux noise. Ideally, the authors should measure the flux-dispersion for the second transition or at least present theoretically expected behavior.

Our simulations for an IST qubit with characteristic energies of $E_j=30$, $E_c=0.16$ and $E_I=0.5$, which is close to the device A parameters in the paper, shows an exponential behavior in flux dispersion as a function of the number of levels similar to charge dispersion of the transmon as shown in the following figure below. Therefore we removed this claim from the paper.

Having said that, we still believe that the IST might be suitable for qudit experiments since the flux noise amplitude is generally smaller compared to the observed charge noise, it is easier to correct for due to its spectral dependence and because the flux sweet spot would be the same for all levels.

6) Fig. 3, caption: I would modify “Higher inductance leads to smaller phase confinement...” to “Higher inductance leads to weaker phase confinement..”

This has been corrected as suggested.

7) Page 4, top-left paragraph: Change “flat vs. flux” to “flat with respect to flux”.

This has been corrected as suggested.

8) Page 4, Eq. (2): The second-order derivative of ω_{p01} with respect to flux was not shown. Maybe add the expression to Methods or to supplement.

This has been added to the methods (Eq. 16) as suggested.

9) Page 4, Table I: Showing Hahn echo decay constants would be useful to understand the spectral content of the flux-noise.

We performed the Hahn echo experiment for device C biased at $\varphi_{\text{ext}} = 0$ with one to three pi-pulses in the sequence. We did not observe an improvement on the measured dephasing times, suggesting that we are dealing with white noise or at least the noise contains high frequency components. We believe that this supports our assessment that the T2 at the flux sweet spot is limited by other noise sources such as photon shot noise due to insufficient shielding and thermalization of the microwave resonator.

We added this information to the caption of Table I: “Echo experiments on device C did not improve the T2 time, which suggests that low frequency flux noise is not the dominant limitation at zero flux.”

10) Page 4, bottom-right paragraph: Change κ to $\kappa/2\pi$ for accuracy.

This has been corrected as suggested.

11) Fig. 5, caption: Change the order either in the sentence “show a fit to the plasmon qubit and readout resonator” or the figures (c) and (d).

As suggested we changed the order in the sentence.

12) Fig. 5b: Why does the x-axis start from 300 mV here? Is the current used as a heat source?

The measurement shown in Fig. 5b is periodic with the bias voltage source within (-1,1) volts, the range (300mv, 400mv) was chosen simply out of convenience. The temperature of the fridge is controlled separately via a factory installed heater inside the fridge. The output voltage of the bias source translates to the current applied to the bias coil with $10 \mu\text{A/V}$, therefore 300 mV corresponds to $3 \mu\text{A}$. The bias coil is made out of superconducting wire and the resistance of the contacts to the coil are in the Ohm range, so the heat generated by the bias voltage was not noticeable.

In the caption of the Fig. 5b we added : “The spectrum continues periodically outside of the shown range.”

13) Page 5, right column: It was not clear what the authors meant by “This is extreme case ... control schemes”. A little more elaboration would be useful.

This point becomes clear in the newly added section “Preparation and decay of long-lived fluxon states” and we therefore changed the sentence to “This is an extreme case for the expected T1 protection of the flux states in this limit [49] which we explore in detail in the final subsection.”.

14) Fig. 6, caption: Given the base temperature of 7 mK and a strong effort to thermalize, it is unusual to have a bath temperature of 108 mK. Some discussion on how the authors arrived at this number would be useful.

We agree with the referee that a bath temperature of >100 mK is surprising. We checked the calculations and noticed that the value of χ , representing the qubits dispersive shift, was entered by mistake as $\frac{\chi}{2}$ in the dephasing rate induced by photon shot noise expression (Eq. 4 in Rigetti, Chad, et al. *Physical Review B* 86.10 (2012): 100506). Correcting this mistake we found the new values for thermal populations to be $n_{\text{th}} = 0.011$ and 0.004 for device A and C respectively. Using the new value for the Bose-Einstein statistics, we arrive at a bath temperature of 90 - 100 mK.

We have corrected these values in the manuscript.

We also checked the possibility of other limitations such as:

- 1) Second order flux noise: $\Gamma = \frac{\partial^2 \omega}{\partial \Phi^2} A^2 = \frac{8\pi^2 \sqrt{8EjEc}}{\hbar \text{bar} (2Ej/EI)^2 \Phi_0^2} A^2$, Which in our case it is not flux dependant anymore since $\omega \propto \Phi^2$ and act the same way as thermal photon shot noise and 2T1 limit. By substituting A as $98 \mu\Phi_0$ one would get a T2 limit of 34 ms.
- 2) Critical current noise which in our case is calculated to be in the order of 3 ms, which is clearly not the limiting factor for the T2 times also.

We believe better thermalization and using more isolators in the readout chains as well as removing the second cavity port could be a way to lower this temperature and check whether the source of the T2 limitations is in fact photon shot noise or not.

15) Page 6, bottom-left paragraph: “Combining a set of independent flux sweeps...” – How are the independent sweeps determined, different starting points in flux or temperature?

The sweeps have fixed frequency and external magnetic field range and all of them are performed at the base temperature of 7 mK. We changed the sentence to clarify this point: “Combining a set of independent flux sweep measurements (with fixed frequency and external flux range), all conducted at the base temperature of 7 mK, in one plot yields”

16) Page 6, bottom-right paragraph: Refer to figure 3 while mentioning $|p1\rangle$ as it is not defined earlier.

We changed the sentence to “...The energy relaxation of the $|p1\rangle$ state (as defined in Fig.3a)....”

17) Page 7, left column: How is the flux-noise amplitude calculated to be $98 \mu\Phi_0$?

By fitting the Eq. 3 to the T2 data with respect to external flux of device C with weak flux noise protection as shown in Fig. 6b. We added a sentence to make this point more clear in page 7 after equation 3: “ The reported number for the flux noise amplitude is obtained by fitting Eq. 3 to the measured external flux dependent T2 time shown in Fig. 6b.”

18) Page 9, after Eq. (13): Typo, the energy difference between first and second excited should have $+2E_C$ in the expression.

This has been corrected as suggested.

19) Page 9, Eq. (15): Change the equality to approximately.

This has been corrected as suggested.

20) Page 9, Device fabrication: The length of the geometric superinductor would be an important number to mention. Since the performance of the device depends very strongly on the fabrication, it would be great if the key steps are shown with a figure along with important precautions taken.

Was there a particular reason to use the air-bridges, as one could modify the geometry to a simple meander and avoid additional fabrication steps?

The length of the geometric superinductor used for device A in the paper is 14 mm. To achieve the same inductance (300 nH) we would need a much longer meander, which is expected to result in significantly lower self resonance frequencies. We would like to note that once the fabrication recipe was conceived and fine-tuned the yield of such air bridges is close to 100% and while it may seem complex and difficult, in practice it is reliable for wire width of at least 200 nm.

We added the length of the coil inductor to the caption of the Fig. 4 and to make the fabrication steps more clear we added a fabrication figure to the methods section.

21) A full circuit diagram including different attenuators and filters used in the experiment would be beneficial for the readers.

As suggested we have added a detailed measurement setup figure to the methods section.

Additional comments:

22) Fig 3b: The colors of traces corresponding to EL = 2 and 0.25 look very similar. Changing the colors to improve contrast would be great.

We have changed the colors as requested.

23) Fig 3d: The discontinuities in the inset look odd.

We added the following explanation to the caption of Fig.3 to clarify this point: “The discontinuities are the result of fluxon states crossing with the plasmon level where the numerical algorithm fails to follow the plasmon state reliably.”

24) Page 4, Table I: Since the authors made a comparison with transmons several times, it would be nice to have all the coherence and dispersion properties for the ω_{p12} transition as well.

Due to time and setup availability limitations we were not able to experimentally back out these numbers for the higher levels. Also, we have removed the claim about the usefulness of higher levels for qudit implementations which should make it somewhat less relevant.

Nevertheless, below in the table we list the expected coherence properties (calculated at $\varphi_e = 0$) of the ω_{p12} transition. Here, the dispersion of the ω_{p12} was extracted from the scQubit simulation library. For relaxation values we employ Fermi's golden rule by using the effective quality factor obtained from the fit for ω_{p01} and replacing the matrix element of ω_{p12} . Finally, the T2 values are calculated using: $\frac{1}{T_2} = \frac{1}{2T_{1p12}} + \Gamma_{th}$, where the T_{1p12} is the expected relaxation time of the ω_{p12} and the Γ_{th} is the effective photon shot noise (we used the thermal occupation obtained from the fit to T2 values of ω_{p01}).

ω_{p12} expected coherence properties	Dispersion [MHz]	Expected T1 [μ s]	Expected T2 [μ s]
Device A	6.4	7.9	10.22
Device B	7.0	10.9	17.7
Device C	48.3	8.9	14.6

Since these values were not measured and confirmed experimentally (due to time limitations) we decided not to add these predicted values to Table I. We ask for the reviewer's understanding regarding our limitations on this point.

Reviewer 3:

The article “A superconducting qubit with noise-insensitive plasmon levels and decay-protected fluxon states” by F. Hassani, et al. describes the theory and implementation of an inductively shunted transmon. The inductively shunted transmon regime is defined as a special type of rf-SQUID for which $E_J(\text{Josephson energy}) \gg EC(\text{charging energy})$ and $E_J \gg EL(\text{inductive energy})$. It can be seen as the first transition of a heavy fluxonium (Earnest PRL 2018) at its upper sweet spot with a very large Josephson energy or as a transmon with a large inductive shunt. In this regime, the first plasmon transition resonates near the plasma frequency $\sqrt{8 EJ EC}$ regardless of the applied external magnetic flux. This property results in a protection against flux noise in addition to the built-in charge noise protection provided by the large inductance shunting the Josephson junction.

The idea of an inductively shunted transmon is not new. As it is stated in the introduction of the paper, it was previously proposed, for instance, to create a new readout coupling scheme (Richer PRB 2017) or to prevent instability with large off-resonant pump amplitude (Verney PRApplied 2019). However, as far as I am aware, the parameter regime studied here has no experimental counterpart in the literature. The topic is relevant and timely as there are significant efforts in the superconducting circuit community aimed at creating new kind of scalable circuits with potential to outperform the widespread transmon circuit (see e.g. Nguyen PRL 2018, Pechenezhskiy Nature 2020, Gyenis PRX Quantum 2021, etc) or to use alternative circuit elements to improve existing circuits (see e.g. Grunhaupt Nature Materials 2019, Peruzzo PRApplied 2020, Place Nature Communications 2021, etc).

The paper is overall well written and sounds very accurate scientifically. The fabrication of the device is impressive and quite unique. All the theory and experimental details are clearly provided in the main text or in the methods. The new qubit regime realized here will certainly trigger a lot

of excitement in the community as it might hold the solution to several technological challenges faced by the community. However, there are several issues and questions that need to be fully addressed before one can consider the paper for publication in a journal like Nature Communications.

We thank the referee for the positive evaluation and the fruitful questions, suggestions and corrections. We did our best to fully address all comments below.

Science Questions

1) Could the authors comment on the stability of the circuit? The authors observe fluxon tunneling on time scales of the order of minutes. This is surely quite impressive and it demonstrates once more the validity of Fermi's golden rule for the rf SQUID. However, thermal excitations of the diamond shape transitions seem unavoidable close to $\phi_{\text{ext}} = \pi$. This is both due to the low frequency of the transition and to the fact that the phase matrix element of this transition becomes maximal at this particular flux point. Will such excitations compromise the stability of the circuit used as a qubit? How is the frequency of the IST transition affected by the tunneling of a fluxon? I think it would be good to clarify the role of these transitions when considering the circuit to store and manipulate quantum information.

We did not observe stability degradation due to the high matrix element at exactly half flux.

The first reason is that the numerically calculated hybridization gap between neighboring wells for the IST qubit device A is only about $\Delta_{\text{ge}} \approx 200$ μHz . It is true that this value is well below the energy of the environment temperature, which is approximately 400 MHz. However, as is evident from the spectrum shown in Fig 5.a at 7 mK, the fluxon states are not in a mixed state (the phase particle is either in one well or another). We believe the reason is that the time dynamics of any transition between the two fluxon states approximately occurs at a timescale of order $1/\Delta_{\text{ge}} \approx 5000$ seconds, which allows long measurements of on average that time.

This is confirmed by the time-domain measurement results added to the paper in the new subsection "Preparation and decay of long-lived fluxon states" showing decay times of 3.6 hours close to half flux.

The second reason is that in the IST limit the range where the flux matrix element between the two wells increases turns into an almost singular point. This is shown in the left image below where at $(0.5 - 10^{-6})\Phi_0$, the matrix element is $< 10^{-8}$. The corresponding current for Φ_0 in our setup is 300 nA and the minimal resolution of our current source is 100 pA. With that said, in our current setup any sweep across the half flux point on a realistic time scale results in a nonadiabatic transition that forces the state to stay in one well as shown in the right image below.

When the phase of the IST qubit state tunnels this results in a frequency shift of the plasmon transitions. The latter varies significantly depending on the external flux bias and ranges from 0 (at exactly half flux) to ~ 10 MHz at zero flux (for device A). This makes the plasmons a great tool to read out the fluxon state (except if tuned exactly to half flux) and we use this for a fast and continuous monitoring of the fluxon states in the newly added section.

2) The authors say that 'high fidelity excited state preparation is possible' when talking about fluxons. What does high-fidelity mean here? How is the fidelity estimated? How is the excited state prepared? I think this sentence needs to be clarified.

In the new manuscript section mentioned above we show that in the presence of a certain high number of photons in the cavity the fluxon states can be excited with high probability. We calibrate a microwave pulse applied at the cavity frequency to fully excite the phase particle to the neighboring higher energy well as shown in the new Fig. 8a in the paper. Using the calibrated excited/reset pulse we perform a state preparation test by sequentially exciting and resetting (ground state preparation) the fluxon state and counting the errors. The following figure shows a single run of the preparation experiment with an error rate of roughly 3%

We have to mention that this precision is only achievable at certain external flux biases - at least without the use of sophisticated pulse shaping techniques. Very close to half flux the strong pump usually prepares a mixture of ground and excited fluxon states. We are still in the process of understanding the physical mechanisms that enable this state preparation scheme and we hope to be able to report more in the near future.

We have added new data to the paper showing decay rates for fluxon states at three different external flux biases with an explanation on how we prepare a fluxon excitation. Since the exact power and maximum achievable fidelity of this method is quite dependent on the external flux as well as the device parameters (similar to the ionization simulation results for transmons) we introduced an active initialization sequence (Figure 8), which works for all flux values and devices with 100% fidelity.

3) Is there a limitation on the minimal frequency of the IST for a given inductance ? It seems that to fulfill the IST condition, the qubit frequency has to be relatively large (typically 5 GHz or more) or that the inductance has to be increased significantly. The authors should maybe comment on the order of magnitude of frequency and coupling strength that can be achieved with such a circuit.

It is a very interesting suggestion to lower the frequency of this qubit type since this could result in significantly longer energy relaxation times of the plasmon states and we in fact plan to explore this direction. The limitation here is the anharmonicity of the qubit. To lower the frequency of the circuit the more one needs to decrease the Josephson energy. This results in weaker confinement of the phase particle, which has to be compensated by increasing its mass (higher shunt capacitance). Since the anharmonicity of the IST in the limit of high E_J/E_I is somewhat similar to the transmon, higher capacitance will decrease the qubit anharmonicity.

Nevertheless, based on numerical simulations it seems that the IST qubit can potentially be designed to operate at lower frequencies than transmons for equal anharmonicities since the dispersion of the IST qubit is sensitive to flux rather than charge and is controlled by E_J/E_I . For

example, fixing the anharmonicity to 100 MHz, it is still possible to make IST qubits with frequencies as low as 1.3 GHz with $E_J=2.5$ GHz and $E_I=0.1$ GHz with a band dispersion as low as 10 MHz as shown below.

In terms of coupling strength, the current design of the IST qubit closely follows the transmon design (capacitive coupling through an antenna pads to a 3D copper cavity) and offers no difference or (dis)advantage. However in the future, the IST could also be designed to be coupled by mutual inductance to a coil resonator as in Ref. Peruzzo, Matilda, et al. *PRX Quantum* 2.4 (2021): 040341. Since the coil resonator can be designed to have high impedance and therefore high zero point voltage fluctuations, potentially one could achieve new limits in coupling strength.

4) Readability and message

Overall, I think that the impact of the paper would be greatly improved if it was more targeted towards the new results presented by the authors. The paper starts with a general review of the rf SQUID properties. In particular, the comparison between the different regime (inductively shunted transmon, transmon, fluxonium, Blochonium) takes the first three figures with very long captions as well as the first three pages of the paper. The actual device – which is the topic of the paper – is only introduced in Figure 4. While I understand the motivation of the authors, I would suggest presenting the IST directly without explaining again the spectrum of the fluxonium and the transmon (Fig2 a,b,e Fig 3 a lower panel), which is usually assumed to be known in transmon/fluxonium papers.

We agree with the referee that this section could be more concise, however we have also received equally positive feedback about the theory section (for example referee 2). At the same time, we wish for the paper to be as accessible to non-experts as well, and for that we believe it is very important to clarify the relationships to existing devices to understand the origin of the new properties. Finally, with the two additional sections on fluxon time dependence and high power readout there is now a better balance between conceptual figures and experimental results.

5) The authors should avoid using jargon such as “circuit’s desire”, “plasmonic character”, “we solve the Hamiltonian”, “dissipative leakage”, “decompactifies the phase”. I think that removing such expressions would avoid any ambiguity.

We agree and made the following changes to the manuscript to remove the possible ambiguities:

- Changed the "...the size of this splitting is a measure of the circuit's desire to favor phase slip over charge tunneling" to "...the size of this splitting is a measure of the inter-well coupling...".
- Changed "... on the other hand the circuit is dominated by the plasmonic character. " to "... on the other hand the circuit is characterized by plasmon transitions."
- Replaced "we solve the Hamiltonian" with "we analyze the Hamiltonian"
- Removed "dissipative leakage" and changed it to "leakage"
- Changed "The large inductive shunt of the IST decompactifies the phase of the transmon and localizes continuous qubit wave functions in wells with discrete flux number." to "The large inductive shunt of the IST decompactifies the phase of the transmon meaning that it localizes continuous qubit wave functions in wells with discrete flux number."

The authors make some claims which are not well supported by experiments yet and which dilute the message of the paper without adding much to the content. See my comments below.

6) The authors claim that their approach "offers a new path forward towards improved qubit encodings in single and multi-mode circuits" in the abstract. I don't see how the approach is new. The parameter regime of the circuit is new but the approach consisting in controlling matrix elements and sensitivity to noise sources with circuit parameters and design is certainly not new (see e.g. Lin PRL 2018, Earnest PRL 2018).

Here we were referring to dynamically (i.e real-time - on the order of the gate time) controlled matrix elements that would be able to take full advantage of the fluxon protection while at the same time allowing for unprotected but very fast gates (see also answer 6 to reviewer 1). Experimental work is ongoing and a theory proposal is currently in preparation. **We do understand the comment of the reviewer though as this is not clear and we have therefore modified the sentence in the abstract to: "In the future, fast time-domain control of the transition matrix elements could offer a new path forward to also achieve full qubit control in the decay-protected fluxon basis."**

7) The authors claim that the circuit can be used as a qudit as higher levels are also protected from charge noise. But does the flux noise protection decrease with level number ? I find this claim very hand-wavy since the lifetime and coherence time data of higher levels of the circuit are not presented in the paper.

As the reviewer suspects we found that indeed the flux dispersion is increasing quickly for higher levels. For a detailed answer including simulation results of flux dispersion and expected coherence times we would like to point the reviewer to our answers to questions 5 and 24 of reviewer 2. **Even though we believe that there might still be some advantages to realize qudits, we admit that they are hand-wavy and we therefore removed this claim from the paper altogether.**

8) The authors claim that their circuit will be compatible with large drive amplitude for high-power readout and parametric stabilization of bosonic codes. Indeed, some studies anticipate that circuits with a bounded potential are more robust to large drive amplitudes. However, as far as I

am aware, this claim is not yet supported by experimental evidence. There are both experimental observation showing a very poor Quantum Non-Demolition character of the readout of fluxonium (see e.g. Ficheux PRX 2021) and relatively good performances (see Gusenkova PRApplied 2021). There is even no clear numerical evidence that this is the case. The paper cited by the author (Verney PRB 2019) suggests that it is the case but it does not access the transient dynamics of the field ramping up in the cavity, which is crucial to understand the process (see arXiv:2203.11235). The claim is also beyond the scope of the paper.

We have conducted additional experiments and added a new section with new data regarding the high power readout of the plasmon state of the IST qubit to the main text of the paper. We show that continuous single shot qubit readout with fidelities and QND-ness above 90% is possible in the IST device, which was not specifically optimized for this experiment in terms of κ and χ . Importantly this is achieved with ~ 600 readout photons and without a JPA.

We note that this photon number is above the deterministic fluxon excited state preparation photon number of around 75, so this indicates that there exists a resonance condition that triggers a quantum tunneling event but for higher powers it is not fulfilled and does not lead to ionization either (e.g. due to the inductive shunt) and thus the measurement remains QND. We have added this explanation to the paper.

We would argue that these results therefore indicate that the inductive shunt is in fact effective to extend the limit of increasing measurement power before losing the QND-ness and fidelity in continuous single shot measurements. However, we now also point out explicitly that another important ingredient could be the exceptional linearity of the geometric superinductor compared to Josephson junction chains. We hope our results motivate further investigation with simulations such as "Shillito, Ross, et al. "Dynamics of transmon ionization." *Physical Review Applied* 18.3 (2022): 034031." for the IST to better understand the role of the inductive shunt.

For a more detailed answer regarding the role of the potential confinement we would like to refer to question 4 of reviewer 1.

Other issues

9) The authors use the terminology 'sweet-spot everywhere' in the abstract. I find it disturbing since a sweet spot has a well established mathematical definition (see Gyenis PRX Quantum 2021) i.e. that the first order derivative of the energy with respect to a given parameter exactly cancels. I understand what the authors meant: the flux sensitivity is strongly suppressed by the choice of parameters as given by Eq. (2). But I would avoid calling it a sweet spot away from integer and half integer flux points.

We borrowed this terminology from transmon where, although the band dispersion with respect to charge offset is exponentially suppressed, it still has a finite value and is not zero. The terminology was used both in the initial transmon results (see e.g. <https://doi.org/10.1007/s11128-009-0100-6>) but also specifically in the mentioned Gyenis PRX Quantum 2021:

[Fig. 5(a)]. A consequence of this delocalization over charge states is that the qubit energy levels are now exponentially flat at all bias points. For charge modes, this “sweet-spot-everywhere” condition is the transmon regime [15,70]. In practice, with typical values of $E_J/E_C \sim$

to convey the meaning that the band dispersion has been systematically flattened up to a point where the first order sensitivity to the respective noise is no longer a limitation for T2 coherence time.

In our opinion this is an intuitive picture that also applies in our case where the sweet spot at zero flux extends over a large enough flux range such that the T2 does not depend on flux anymore. Nevertheless, since this was also criticized by referee 2 - and to avoid any misunderstandings - we have removed the term from the paper.

10) The authors say that their circuit exhibits ‘ultra-small’ matrix elements (caption Fig 2). Small compared to what ? Later on in the main text, it said that the matrix element is in the range $10^{(-13)}$. Are we talking about the charge matrix element as represented in Fig 2 d ? Is that value extracted from numerical diagonalization ? Isn’t it limited by numerical errors ?

Yes the values are shown in Fig. 2d (on a linear scale) and also earlier in Fig. 1 as contour lines (on a dB scale). The value is ultra-small compared to all other RF SQUID type qubits shown in Fig. 1. Specifically it is a billion times lower than the closest one (the heavy fluxonium) where a direct excitation and control of the fluxon state is still possible.

The numerical accuracy of the result, using the ScQubits python library, is highly dependent on the number of basis states used to perform the calculations. In ScQubits this parameter is set using the cutoff entry for any qubit type (here fluxonium). In the following figure we show the charge matrix element of an IST qubit similar to what is shown in Fig. 2d ($E_J=30$ GHz, $E_C=0.17$ GHz, $E_I=0.5$ GHz) calculated with different cutoff limits. Here, the transition (0-1) within the flux range (0.18,0.85) and the transition (0-2) outside of the flux range (0.18,0.85) represents the charge matrix element value for the fluxon transition (as shown in Fig. 2d with the color yellow). By increasing the cutoff limit the charge matrix element converges to what was reported in the paper ($10^{(-13)}$) at the cutoff 150 (further increase of the cutoff no longer changes the result). The accuracy used in all the simulation inside the paper is set to be 160.

We have added this information in the part of the paper where we discuss the fitting procedure in the methods section: “Another important factor for obtaining accurate numerical results for IST matrix elements, specially for flux transitions because of their extremely small values as low as 10^{-13} , is to use a sufficiently high cutoff value for the number of basis functions used to carry out the calculations. In all the simulations shown in the paper such as in Fig.1, 2 and 3 we use a cutoff value of 160 in the scQubits library to achieve accurate results.”

Minor issues

11) I think that the acronym IST is first used without being defined.

We added the definition to the caption of the Fig.1 and abstract.

12) The x-label of Fig 3 d is shifted to the right.

We have corrected this.

13) In Fig 3 a, why are the odd level wave functions mirrored with respect to $\phi=0$ between the transmon and IST ?

We have corrected this.

14) The inset of Fig 3 d is discontinuous, is it because of the avoided crossing with the fluxon lines ?

Yes, the discontinuities are caused by the avoided crossing with the fluxon transitions and we have clarified this in the caption.

15) Typo: Methods, Fitting procedure, "einenenergies"

We have corrected this.

16) Typo: p7 left column last paragraph "it's"

We have corrected this.

17) Caption Fig4 missing space after "d."

We have corrected this.

General remark:

The authors have revised the manuscript substantially and satisfactorily addressed my previous comments. In the new version, they clarify the language to indicate the trade-off between flux tunability and flux noise insensitivity, elucidate the distinction between their qubit and other inductively shunted platforms, and add valuable measurements on readout QND-ness and the lifetimes of fluxon states. They also followed the other reviewers' suggestions and significantly improved the lucidity of the text. The combination of the reported data on the plasmon's flux (in)sensitivity, the fluxon's record lifetimes, and the encouraging readout fidelity will be valuable to the field. I believe that the manuscript is suitable for publication in Nature Communications with some modifications on the new sections. I look forward to the future progress on the inductively-shunted-transmon qubit.

Detailed remarks:

Since the authors have added corrections to the previous version, I hereby include comments on the new sections. I hope this will help verify and substantiate the additional data reported.

Readout of plasmon regime at zero flux:

1. On the readout fidelity, can the authors clarify how they perform state initialization and how it may affect the extracted fidelity?
2. I am surprised the readout fidelity for the excited state is so low. Can the authors comment on the probable cause?
3. I would appreciate a legend on the curves in Fig. 7 (a,b), in addition to the description in the caption.

Measurement of fluxon decay:

4. On the long-lived fluxon initialization, I believe including some data in addition to Fig.8(a), possibly as supplementary, will help describe the story much better. For example: state population vs power and pulse length, at the reported flux points.
5. I find this regime to be even more intriguing than the plasmon regime from the description provided. Can the authors further clarify how the phenomenon can be attributed to a resonance condition? I believe the result contradicts the observation in Gusenкова et al., where the readout QND-ness of the fluxon states were found to be excellent.
6. In general, I find the description in the main text to be handwavy. "Even higher powers", "certain flux values", and "perfect state preparation" do not appear scientifically precise. I suggest referring to Fig. 8(a) in the main text.

7. Can the authors clarify how to confidently measure a preparation error of 3% given their readout fidelity, which I assume to be similar to the reported values zero-flux point? Better yet, can the authors provide related information about the readout QND-ness and fidelities at the relevant flux points, and how they use this to quantify the initialization fidelity?
8. I appreciate the authors' honesty in revealing the non-QND-ness of the IST readout away from zero-flux. I recall that a similar effect and a subsequent reset technique were reported previously for fluxonium biased at a metastable regime (L. Nguyen's thesis, Fig. 6.9) and at half-integer flux (Ficheux et al., Appendix C.3). However, these works do not elucidate the physics behind the effect. It is quite interesting to see the same physics at play even though the parameters are markedly different. Besides the provided simulation results in transmons, do the authors have an insight on this? I suggest clarifying the distinction between the current work and the previously reported initialization data.
9. For the measurement, the authors indicate that they readout the plasmon state in the single-photon limit to ensure QND-ness. However, is there some way to verify this? It is an important detail to validate the measured decay data, since the resonator photons tend to reset the qubit in the excited fluxon state. Ideally, QND-ness information on the three flux points should be sufficient to support the T_r data. If such evidence is unobtainable, I suggest some comments on how it should be done in future experiments.
10. Alternative to the plasmon-assisted measurement, what is the dispersive shift corresponding to the fluxon transition? Is it possible to measure directly? Fig. 5 indicates that it seems possible to distinguish the fluxon states dispersively, so it helps to justify the measurement scheme.
In case the direct dispersive measurement is possible, can it be performed to check for consistency?
11. For Fig. 8 caption, I believe the sequence description should better be included in panel (b) instead of (c).
12. In the summary, please clarify again that the measured QND-ness is expected to be valid at zero flux (unless the authors will report additional data at different flux points as suggest above).

Minor comments:

- I suggest unifying the writing style. For example, I see “readout”, “read out”, and “read-out” in the text.
- Fig. 5 caption: I suggest unifying the notations in the figures and the caption.
- Fig. 8 caption: “plansmon” -> “plasmon”
- In Methods, in addition to the cutoff value, can the authors specify which basis is used for numerical simulation?

Reviewer #2 (Remarks to the Author):

First, I am delighted to see that the authors found a way to prepare and measure the relaxation time of the fluxon states. It is impressive to see such a long T_1 , verifying the action of Fermi's Golden rule. I am satisfied with the responses to the comments made by the other referees and mine. However, I have two more concerns that the authors should address.

1) Fig. 8b – A few important details are missing for the fluxon state preparation and readout. What is the difference (Δ) in transition frequencies of the plasmon states conditioned on the fluxon states? I understand that it has a flux dependence, but the authors can provide the values for the flux points they performed experiments at. What is the pi-pulse length used for plasmonic transition? This Rabi rate should be significantly slower than the frequency difference Δ . For the fluxon readout, are the authors applying plasmon drives corresponding to both $|f_0\rangle$ and $|f_1\rangle$ alternatively? A negative excitation of the plasmon for $|f_0\rangle$ may not guarantee the fluxon state being in $|f_1\rangle$ and needs to be verified by a plasmon transition conditioned on $|f_1\rangle$. If the plasmon is excited for the case $|f_1\rangle$, is another pi-pulse applied to bring the plasmon to its ground state, or simply a waiting period is applied? What is the resonator probe power used for fluxon state preparation? I guess it is around 70-80 photons corresponding to the bottom panel of Fig. 8a but should be explicitly mentioned.

2) Fig. 1 – I missed this point in the first round. Considering the parallel circuit model (as in Fig. 2a inset), EL for both the transmon and CPB should be 0. Thus, both of those should be placed on the right side of the plot at $EJ/EL \rightarrow \infty$. This placement also aligns with the fact that the transmon (single junction) should have no flux sensitivity. I suggest putting a line break on the x-axis and place those on the right vertical axis.

Other comments

1) Eq.(2) – I would use " \approx " instead of " $=$ "

2) Please verify that Eq.(2) and (3) are consistent with each other as there seems to be a factor of 2 discrepancy. Also, it would be better to consistently use either $2EJ/EL$ or EJ/EL throughout the paper.

3) Fig. 7 – The shaded area in 7(a) appears to be not equal to 128 ns as mentioned in d (caption).

4) Page 12, right column – change "The second order derivative can be calculated from Eq. 16" to "The second order derivative can be calculated from Eq. 15"

5) Fig. 3a (minor suggestion) – For the kets, use the same colour as the wavefunction lines for better visualization.

Reviewer #3 (Remarks to the Author):

I would like to thank the author for the effort that they put in replying to the referees and editing their paper. I think that the answers to the referees comment are very detailed and overall satisfying. However, the author decided to significantly change the main text of the paper (it is now at least 20% longer than the initial submission + 2 new figures in the main part of the paper) beyond what was required by the referees in my opinion. These changes require reconsidering entirely the manuscript before publication.

The two new sections added by the author do not sound sufficiently scientifically accurate to me and I am now afraid that they will bring a lot of questions and confusion to readers. I would even suggest removing the discussion about "QNDness" from the paper. I also think that the readability of the new sections are not as good as the rest of the paper. This is very unfortunate. I must say that I find the results of the paper in the initial submission important for the community but, as such, I am forced to still not recommend the paper for publication.

- I strongly disagree with the analysis of the data of Figure 7. The authors calculate a "QND-ness" as $(P_{g,g} + P_{e,e})/2$ where $P_{i,i}$ is the probability that the state i is detected in two successive independent measurements. However, the two successive measurements are "subsequent data points that are 1 μ s apart" during the same readout pulse. They say that the readouts are "independent" because they are separated by more than 1 μ s ($>$ correlation time induced by the filter).

I don't understand how this can be valid. With this analysis, any bifurcation readout scheme that yields a constant signal after bifurcation (see e.g. Reed PRL 2010) would look very QND, while they are not. That is the qubit state cannot be reused right after the readout pulse in a high-power readout measurement (such as the one of Reed et al.) even if there is a very strong time correlation within one measurement record trace.

As such, without a clarification from the authors, I believe that the results presented in Fig7 c are potentially scientifically incorrect. Or more precisely, that the quantity calculated by the author does not quantify the "QND-ness" of the readout process, as claimed in the paper.

-“To quantify the QND-ness we analyze the data starting at $10/\kappa \approx 1.5 \mu\text{s}$, which we consider to be in the steady state response limit of the cavity”.

The authors are ignoring the transient regime of the readout pulse to determine the “QND-ness”. The transient regime is a part of the readout process and ignoring it makes the metric meaningless for practical applications. It obliterates potential transitions during the transient of the cavity field, which is the main limitation in practical applications. Performing two successive strong readout pulses separated by a time $\gg 1/\kappa$ would have clarified everything.

- The authors mention multiple times the use of a Chebyshev filter applied to the measurement records. I do not understand why the authors cannot show the raw data. In the literature, measurement records such as the one presented in Fig 7 a and b are shown without any additional filtering beyond the one of the detection chain (see e.g. Hatridge Science 2013 or Dassonneville PRX 2020). Could the authors consider removing this filter? It raises a lot of questions

(i) Within a single trace, I imagine that successive measurement records for each time bin are now correlated since $1/1\text{MHz}$ is not negligible compared to the binning time of the measurement records shown in Fig 7a,b. What is the smoothing effect on the ‘quantum jump’ in b? Claiming that this trace can be interpreted as a quantum jump after applying such a filter is non trivial to me.

(ii) What is the impact on the noise and the associated error bars for the mean measurement record?

(iii) What is the impact on the 128 ns-long measurement used to determine the fidelity?

(iv) What is the order of the filter?

- “a temporal mode matching function [...] was used”.

I imagine that the data were processed further for the fidelity measurement. Not explaining the data processing (i.e. which function was used, why or the order of the filter) makes the results not very meaningful and irreproducible for other research groups. I am not even able to guess the function since different research groups use different filters. Is it a boxcar linear filter, an exponentially decaying linear filter, “optimal” linear/nonlinear filter, Bayesian filter? That is critical information when discussing readout performances.

-“While these results generally agree with the expectations from steady state simulations presented in [19]”.

Similarly to the previous version of the paper, the authors want to make connections which are not supported by their results. I don’t know why they want or need to say that. The paper by Verney et al. described the simulation of a circuit with parameters that are more than one order of magnitude different from the ones of the authors. For instance in their work $\text{EL} = 0.5 \text{ GHz}$, while for Verney et

al. EL = 14 GHz. I did not find in the work of Verney et al. any discussion about QNDness or readout fidelity that can show an agreement with the results reported by the authors.

Now the claim is clearly there, I am therefore forced to ask the author to show an agreement between their Figure 7 and simulations such as the ones of Verney et al. in the relevant parameter regime of the experiment. In this way, one can see if there is an agreement between the simulation and the value of fidelity, QNDness, and typical photon number reported in Figure 7. Or at least one would be able to judge if the interpretation of Verney et al. is still valid for such a small inductive confinement.

- The active fluxon excited state preparation is a very good idea. But in my understanding, the excitation scheme is not deterministic as stated in the conclusion ? Did the author consider a bichromatic initialization scheme ? What is the success rate of each preparation cycle ? What is the average duration of the total preparation ?

- The authors claim a “perfect state preparation fidelity” of the fluxon with the protocol depicted in Fig. 8b. This must mean that the conditional plasmon excitation as well as its readout (in the single-photon limit) have a 100% fidelity ? That sounds too good to be true ? Similarly, it would be good to explain how the finite plasmon measurement fidelity is taken into account to calibrate the y-axis of the Fig. 8c. Now it seems that it starts perfectly in 1 and ends perfectly in 0, which is only possible with the perfect state preparation fidelity mentioned above ? And the quantity ρ_{f01} reported on the y-axis is an element of a matrix density, I guess. Is it defined somewhere ?

- Could the author comment on the different discretization of the curves in Fig. 8c. ? Is it due to a reduction of the dispersive shift between the plasmon and the resonator or between the fluxon and the plasmon ?

- The authors show impressive lifetimes for the fluxon transition. How to exclude other slow relaxation mechanisms such as the slow environment time relaxation reported <https://arxiv.org/pdf/2204.00499.pdf> ? How can the reader be sure that the curves reported in Fig 8c are not due to slow temperature drifts of the measurement setup ? Are the lifetimes reported here compatible with the wavefunction overlap at different flux points ?

- “based on a 1 MHz Chebyshev filter”. I don’t understand the word “based” here, is it the correct word ?

- "populate 40×10^3 data points in the quadrature histograms" -> is the word "populate" the correct word here ?

Reply to all referees

We thank all three reviewers for taking the time to give us valuable feedback to further improve our work. We acquired new data in an additional cooldown with independent subsequent pulses for QND-ness as well as higher time resolution quantum jumps. As suggested, we also moved the bulk of the read-out material to the methods section and we think this helped to simplify the corresponding discussion in the main text. We emphasize that this did not lead to new qualitative or significant quantitative changes to the previous conclusions. Similarly, we provide substantial additional material about the fluxon state preparation and readout in the methods section. All changes and additions to the main text are highlighted with red color in both documents (except for the new methods sections which are not color coded).

In the following, we respond to all reviewer comments in detail. Answers are in blue and related changes to the manuscript are in red font.

Reviewer 1

General remark:

The authors have revised the manuscript substantially and satisfactorily addressed my previous comments. In the new version, they clarify the language to indicate the trade-off between flux tunability and flux noise insensitivity, elucidate the distinction between their qubit and other inductively shunted platforms, and add valuable measurements on readout QND-ness and the lifetimes of fluxon states. They also followed the other reviewers' suggestions and significantly improved the lucidity of the text. The combination of the reported data on the plasmon's flux (in)sensitivity, the fluxon's record lifetimes, and the encouraging readout fidelity will be valuable to the field. I believe that the manuscript is suitable for publication in Nature Communications with some modifications on the new sections. I look forward to the future progress on the inductively-shunted-transmon qubit.

We thank the reviewer for the positive evaluation.

Detailed remarks:

Since the authors have added corrections to the previous version, I hereby include comments on the new sections. I hope this will help verify and substantiate the additional data reported.

Readout of plasmon regime at zero flux:

1. On the readout fidelity, can the authors clarify how they perform state initialization and how it may affect the extracted fidelity?

The plasmon ground state initialization was implemented by waiting about ten T_1 times after each measurement pulse (200 μs). This number was optimized experimentally by increasing the measurement rate and observing the fidelity of the ground state preparation to saturate.

The preparation of the excited state is performed by applying a calibrated (via a standard Rabi experiment) microwave pulse on resonance with the plasmon transition frequency via the

resonator port. For this we directly gate a microwave generator without any control of the exact pulse shape. We found an optimal fidelity for a 40 ns long pi pulse. We expect that this very simple excitation scheme contributes (or maybe even dominates) the reduced excited state fidelity.

We added the information above to the caption of Fig.7.

2. I am surprised the readout fidelity for the excited state is so low. Can the authors comment on the probable cause?

Up to about 7% infidelity can be attributed to the readout time of 500 ns during which the excited state decays. This is assuming $T_1 = 7 \mu\text{s}$, which is the measured number for this 5th cooldown of this sample due to a consistent T_1 degradation compared to the first cooldowns.

The limited ground state fidelity indicates that there is only a 1.3% chance to be in e.g. a thermally excited plasmon or fluxon state rather than the ground state when the pi pulse tries to excite the qubit. This is shown below for the pulsed high photon number readout where we reach the highest fidelity:

We attribute the remaining part of the infidelity of at least 5% to a limitation of our current setup. We use square digital pulses to gate the RF sources directly rather than using pulse-shape optimized Gaussian or DRAG pulses to excite the qubit (as well as for the readout tone). In addition, the pi pulse is applied via the cavity port rather than directly via an AC drive line that is common in on-chip experiments.

We summarized the above in the main text and added the histogram to Fig.13d in the methods section.

3. I would appreciate a legend on the curves in Fig. 7 (a,b), in addition to the description in the caption.

We performed new quantum jump measurements with higher time resolution and moved the data to the methods section. There we now show the raw data of many traces with higher time resolution in Figure 13 b. The goal was to clarify this as much as possible without adding too much to the main text. Instead we extract QND-ness from two subsequent measurement pulses and report the result of that in the main text (on the request of reviewer 3).

Measurement of fluxon decay:

4. On the long-lived fluxon initialization, I believe including some data in addition to Fig.8(a), possibly as supplementary, will help describe the story much better. For example: state population vs power and pulse length, at the reported flux points.

We added a corresponding section to the methods. It explains the details of the fluxon excitation and reset. Specifically, the new Figure 14 shows the fluxon state dependent plasmon spectroscopy as a function of resonator pump power (and photon number) for the three flux bias points discussed in Fig. 8c.

5. I find this regime to be even more intriguing than the plasmon regime from the description provided. Can the authors further clarify how the phenomenon can be attributed to a resonance condition? I believe the result contradicts the observation in Gusenkova et al., where the readout QND-ness of the fluxon states were found to be excellent.

We are in an active collaboration with theorists to explain the resonance phenomenon used for the excited fluxon initialization. Our current hypothesis is that the resonator photons act as an effective temperature that facilitates tunneling and this scales more strongly compared to what would be expected in a harmonic potential. More importantly, the resonance condition shown most clearly in Fig. 14a (right plot) is very likely due to a combination of Stark shifts (power dependent) and a multi-photon transition that is partially allowed due to the anharmonic potential, thus breaking the usual selection rules. In the specific case shown, the resonance is likely due to 5 plasmon energies, corrected by the power-dependent Stark shift, exactly matching 3 cavity photon energies.

Importantly, the specifics of this phenomenon depend sensitively on the device parameters as well as the flux value (for example in panel b there are multiple resonance starting at lower photon numbers and in panel c there is just a mixture above a certain low photon number). Therefore, we do not think that it necessarily contradicts the results of Gusenkova et al., which works in a very different parameter regime. We also find that a (plasmon) QND readout is possible up to very high photon numbers as long as resonant tunneling at certain photon numbers is avoided. This is a conceptual difference to the transmon case where ionization happens (excitation to unbounded states above the cosine potential).

We plan to work out the details of this tunneling physics in a follow up paper when it is fully understood and would like to avoid adding more material to the current manuscript.

6. In general, I find the description in the main text to be handwavy. “Even higher powers”, “certain flux values”, and “perfect state preparation” do not appear scientifically precise. I suggest referring to Fig. 8(a) in the main text.

Adding the power sweeps (Fig. 14) to the methods allows us to be more specific and we made a number of changes to make the language and contents more precise (highlighted in red in the main text).

7. Can the authors clarify how to confidently measure a preparation error of 3% given their readout fidelity, which I assume to be similar to the reported values zero-flux point? Better yet, can the authors provide related information about the readout QND-ness and fidelities at the relevant flux points, and how they use this to quantify the initialization fidelity?

We emphasize that the plasmon assisted readout of the fluxon state is an averaged measurement as shown in Fig 8.b, therefore the single shot fidelity of the plasmon state does not directly limit the preparation or detection of fluxon states. In fact, we do not use the single shot high power readout of the plasmon levels for detecting the fluxon state. Instead we use a low power readout of the plasmons close to the single photon limit to avoid any unwanted photon assisted fluxon tunneling together with averaging as shown in Fig. 8b. While this readout of the plasmons takes long (100 ms), in comparison to the lifetime of the fluxons it is negligible.

Here is an explicit example of a state preparation test where we sequentially excite and reset (to the ground state) the fluxon state once, and then perform the plasmon readout multiple times and average the result. Counting the errors, we arrive at an error rate of roughly 3%:

This is valid at zero flux and does not work as well closer to flux degeneracy as can be seen in Fig. 14. For those flux values we used an active state preparation scheme as discussed in the manuscript.

8. I appreciate the authors' honesty in revealing the non-QND-ness of the IST readout away from zero-flux. I recall that a similar effect and a subsequent reset technique were reported previously for fluxonium biased at a metastable regime (L. Nguyen's thesis, Fig. 6.9) and at half-integer flux (Ficheux et al., Appendix C.3). However, these works do not elucidate the physics behind the effect. It is quite interesting to see the same physics at play even though the parameters are markedly different. Besides the provided simulation results in transmons, do the authors have an insight on this? I suggest clarifying the distinction between the current work and the previously reported initialization data.

Even though we have not directly tested the high power plasmon readout close to half flux, the new Figure 14 clearly suggests that indeed, this flux range (panels c and d) is not suitable since a growing range of resonator powers lead to photon assisted tunneling to another fluxon state.

We thank the reviewer for bringing the relevant results from L. Nguyen's and Ficheux et al. to our attention. **We added these two citations to the corresponding sections in the main text.**

With regards to the relationship to our work we have to say that the potential and eigenenergies of the heavy fluxonium are certainly similar to the IST qubit. The main difference is the up to 10^{10}

times lower transition matrix element for fluxon transitions as well as the number of single well confined plasmon states – as pointed out in the paper. This allows us to study the effect directly with conventional spectroscopy.

These differences in parameter regimes also affect many of the specifics of the photon assisted tunneling physics that we already described in our answer to question 5. Understanding it better might allow to design a circuit that enables high power QND readout at specific photon numbers by avoiding resonance conditions also closer to half flux.

We are in the process to follow up on the physics with experiments and theory that is quite rich, but at the same time does not need to rely on dissipation or extremely heavy numerics for qualitative insights.

9. For the measurement, the authors indicate that they readout the plasmon state in the single-photon limit to ensure QND-ness. However, is there some way to verify this? It is an important detail to validate the measured decay data, since the resonator photons tend to reset the qubit in the excited fluxon state. Ideally, QND-ness information on the three flux points should be sufficient to support the T_r data. If such evidence is unobtainable, I suggest some comments on how it should be done in future experiments.

The reported QND-ness, e.g. in Fig. 7 refers to the high-power single-shot plasmon readout. This is not directly relevant for the QND-ness of the fluxon state readout because the latter is based on an average of many low power plasmon measurements. However, also this fluxon readout can be non-QND of course if fluxon transitions are induced by the plasmon readout tone as the referee points out correctly.

The role of the readout photon number on the fluxon state can be seen in the new Fig. 13d (bottom), where a third maxima in the single-shot plasmon readout histograms appears, which we can attribute to induced fluxon transitions. This starts to be visible for around 2500 resonator photons. If a lower number is used (1400 photons) its contribution is $< 1.3\%$ infidelity to the plasmon fidelity. This is consistent with the power sweeps of average measurements shown in Fig. 14 at 3 different flux points down to as low as around 1 average intra-cavity photon. These three plots directly show that for the low power readout conducted at -45 dBm, i.e. about 2 resonator photons, there is no visible probability to excite a fluxon.

The main arguments contradicting the hypothesis that the plasmon readout causes the long observed fluxon lifetimes by exciting the fluxons are the following:

- The fluxon lifetime is strongly flux bias dependent as expected for the simple tunneling model (quantitative investigations will follow), while there is no such dependence expected for the readout and also not visible in Fig. 14 at the level of 2 readout photons.
- We perform a readout only every 30 seconds so the probability would need to be high to make a difference and again this is not expected at such low photon numbers (and not visible in Fig. 14).
- Furthermore, if the readout would excite the fluxons then we would not observe a complete relaxation to the f_0 state for very long timescales. To that end, it should be said that each fluxon state corresponds to a unique resolvable dispersive shift of the plasmon frequency and thus to a specific and directly resolvable dispersive shift of the resonator. At long time scales the measured resonator transmission corresponds exactly to the expected one for the $|f_0\rangle$ state.

In the future, to further corroborate whether the measurement could have an influence on the observed long relaxation rates, we will bias the qubit at different flux values and directly study the effect of the plasmon measurement probe power on the resulting decay rate.

10. Alternative to the plasmon-assisted measurement, what is the dispersive shift corresponding to the fluxon transition? Is it possible to measure directly? Fig. 5 indicates that it seems possible to distinguish the fluxon states dispersively, so it helps to justify the measurement scheme. In case the direct dispersive measurement is possible, can it be performed to check for consistency?

The calculated matrix elements of the fluxon transitions are on the order of $\langle f_0 | n | f_1 \rangle = 10^{-13}$, which directly decouples these transitions from the resonator and consequently the dispersive shift of the fluxon transition goes to zero. In Fig. 5 we plot the spectroscopy of the plasmon states with respect to flux and temperature and we did not observe any direct trace of the fluxon transition within the capability of measurement setup.

11. For Fig. 8 caption, I believe the sequence description should better be included in panel (b) instead of (c).

This is already the case. Caption c is just mentioned inside caption b.

12. In the summary, please clarify again that the measured QND-ness is expected to be valid at zero flux (unless the authors will report additional data at different flux points as suggest above).

We clarified this point by adding in the conclusion “The reported observation of quantum jumps without the need for a near quantum-limited amplifier via a high photon number qubit readout with high QND-ness and fidelity above 90%, **as demonstrated at zero flux**, provides supporting evidence for this hypothesis.”

Minor comments:

- I suggest unifying the writing style. For example, I see “readout”, “read out”, and “read-out” in the text.

We now use “readout” consistently.

- Fig. 5 caption: I suggest unifying the notations in the figures and the caption.

We did not find relevant discrepancies – at least not in the latest version of the figures – but we would be more than happy to make changes once we know the specifics.

- Fig. 8 caption: “plansmon” -> “plasmon”

This is corrected now.

- In Methods, in addition to the cutoff value, can the authors specify which basis is used for numerical simulation?

We added the information to the methods section: “...is to use a sufficiently high cutoff value for the number of **harmonic oscillator basis functions** used to carry out the calculations.”

Reviewer 2

First, I am delighted to see that the authors found a way to prepare and measure the relaxation time of the fluxon states. It is impressive to see such a long T_1 , verifying the action of Fermi's Golden rule. I am satisfied with the responses to the comments made by the other referees and mine. However, I have two more concerns that the authors should address.

1) Fig. 8b – A few important details are missing for the fluxon state preparation and readout. What is the difference (Δ) in transition frequencies of the plasmon states conditioned on the fluxon states? I understand that it has a flux dependence, but the authors can provide the values for the flux points they performed experiments at.

The difference in plasmon transition frequency starts with a minimum of 6 MHz for the closest point to half flux (purple color in Fig. 8c), it increases to 26 MHz midway to zero flux (yellow color) and reaches its maximum of 150 MHz for zero flux (green color). These values are directly visible in the new methods section in Fig. 14, showing the two tone plasmon spectroscopy versus resonator pump power for the three flux bias values, where we reported the fluxon relaxation rates.

What is the pi-pulse length used for plasmonic transition? This Rabi rate should be significantly slower than the frequency difference Δ .

The pi-pulse power and length were chosen in a way to achieve a Rabi frequency < 1 MHz and added this information to the main text (we used a 3 μ s long excitation pulse).

For the fluxon readout, are the authors applying plasmon drives corresponding to both $|f_0\rangle$ and $|f_1\rangle$ alternatively? A negative excitation of the plasmon for $|f_0\rangle$ may not guarantee the fluxon state being in $|f_1\rangle$ and needs to be verified by a plasmon transition conditioned on $|f_1\rangle$.

For the measurements that we show here we only applied one such pulse. In the case of zero flux (green in Fig. 8c) it was applied to the plasmon frequency corresponding to the $|f_0\rangle$ state. In the other two cases it was applied to the one corresponding to the $|f_1\rangle$ state. The choice was based on getting the maximum signal to noise ratio of the averaged measurement for the two states of interest. We have added this information to the main text.

We agree that a conditioned measurement would be a further improvement and we plan to do this in the future. Nevertheless, in this case it was not strictly necessary since not only the plasmon states can be resolved in the spectrum but also the fluxon states up to at least $|f_2\rangle$ lead to a unique dispersive resonator shift with a unique read out amplitude for a well-chosen readout frequency. Therefore, the readout signal goes from a well understood level to another well understood level, each corresponding to a specific state of the system.

If the plasmon is excited for the case $|f_1\rangle$, is another pi-pulse applied to bring the plasmon to its ground state, or simply a waiting period is applied?

We wait for approximately ten times the plasmon decay rate to reset the plasmon state.

What is the resonator probe power used for fluxon state preparation? I guess it is around 70-80 photons corresponding to the bottom panel of Fig. 8a but should be explicitly mentioned.

We use a resonator pump power corresponding to about 72 photons to prepare the first excited fluxon state with high probability. For the readout via the plasmon frequency we use around 2 photons only. We have added a new section in the Methods and Fig. 14 to clarify the details of state preparation.

2) Fig. 1 – I missed this point in the first round. Considering the parallel circuit model (as in Fig. 2a inset), EL for both the transmon and CPB should be 0. Thus, both of those should be placed on the right side of the plot at EJ/EL \rightarrow infinity. This placement also aligns with the fact that the transmon (single junction) should have no flux sensitivity. I suggest putting a line break on the x-axis and place those on the right vertical axis.

As suggested, we have detached the charge qubit axis from the EJ/EL axis in Fig. 1.

Other comments

1) Eq.(2) – I would use " \approx " instead of " $=$ "

This has been changed.

2) Please verify that Eq.(2) and (3) are consistent with each other as there seems to be a factor of 2 discrepancy. Also, it would be better to consistently use either $2EJ/EL$ or EJ/EL throughout the paper.

We thank the reviewer for bringing this into attention. The equations are now consistent and written in the form of $2EJ/EL$. Figures 3d and 9 were also corrected accordingly.

3) Fig. 7 – The shaded area in 7(a) appears to be not equal to 128 ns as mentioned in d (caption).

In this revision, we followed the suggestion of reviewer 3 to apply two consecutive readout pulses to calculate QND-ness in the main text. Quantum jumps and the QND-ness of the quantum jumps are presented in Methods. Instead of the old Fig 7 a&b we are now presenting the raw data of multiple traces in Fig 13b, which shows the quantum jumps more clearly.

4) Page 12, right column – change "The second order derivative can be calculated from Eq. 16" to "The second order derivative can be calculated from Eq. 15"

This has been fixed.

5) Fig. 3a (minor suggestion) – For the kets, use the same colour as the wavefunction lines for better visualization.

This has been implemented.

Reviewer 3

I would like to thank the author for the effort that they put in replying to the referees and editing their paper. I think that the answers to the referees comment are very detailed and overall satisfying. However, the author decided to significantly change the main text of the paper (it is now at least 20% longer than the initial submission + 2 new figures in the main part of the paper)

beyond what was required by the referees in my opinion. These changes require reconsidering entirely the manuscript before publication.

The two new sections added by the author do not sound sufficiently scientifically accurate to me and I am now afraid that they will bring a lot of questions and confusion to readers. I would even suggest removing the discussion about "QNDness" from the paper. I also think that the readability of the new sections are not as good as the rest of the paper. This is very unfortunate. I must say that I find the results of the paper in the initial submission important for the community but, as such, I am forced to still not recommend the paper for publication.

- I strongly disagree with the analysis of the data of Figure 7. The authors calculate a "QND-ness" as $(P_{g,g} + P_{e,e})/2$ where $P_{i,i}$ is the probability that the state i is detected in two successive independent measurements. However, the two successive measurements are "subsequent data points that are 1 μ s apart" during the same readout pulse. They say that the readouts are "independent" because they are separated by more than 1 μ s ($>$ correlation time induced by the filter).

I don't understand how this can be valid. With this analysis, any bifurcation readout scheme that yields a constant signal after bifurcation (see e.g. Reed PRL 2010) would look very QND, while they are not. That is the qubit state cannot be reused right after the readout pulse in a high-power readout measurement (such as the one of Reed et al.) even if there is a very strong time correlation within one measurement record trace.

As such, without a clarification from the authors, I believe that the results presented in Fig7 c are potentially scientifically incorrect. Or more precisely, that the quantity calculated by the author does not quantify the "QND-ness" of the readout process, as claimed in the paper.

We have performed another cooldown of this device (the 5th) to gather data for the QND character with two independent and subsequent measurement pulses. The result of it is presented in Figure 7. While the main text is more compact now, we also provide more details in the Methods, specifically Figure 13. It outlines the histogram data, the raw data quantum jump traces with and without a preceding excitation pulse, as well as pulse sequences used for each type of measurement.

We have also tried to make an effort to clean up the main text and at the same time be more accurate. Specifically, we fixed an inconsistency how we previously reported the measurement time. The new measurements are all reported for 500 ns readout pulses and in the case of quantum jumps for a 500 ns readout time/filter.

We would like to emphasize that our previous analysis of QND-ness based on a continuous measurement (now reported in Fig. 13 c) is not new (we follow Refs 65, 66, 67) and the maximum value is consistent with the independent readout that we now present in the main text. Note however that the photon number values do not agree since we calibrated them for a continuous readout tone, i.e. they are strictly speaking only valid for the quantum jump measurements as also stated in the main text.

Compared to our measurements, the high-power readout firstly showed by Reed PRL 2010 is conducted at significantly higher powers right where the cavity frequency shifts to the bare cavity frequency, at which point the output of the cavity does not contain any information regarding the quantum trajectories and quantum jumps.

-“To quantify the QND-ness we analyze the data starting at $10/\kappa \approx 1.5 \mu$ s, which we consider to be in the steady state response limit of the cavity”.

The author are ignoring the transient regime of the readout pulse to determine the “QND-ness”. The transient regime is a part of the readout process and ignoring it makes the metric meaningless for practical applications. It obliterates potential transitions during the transient of the cavity field, which is the main limitation in practical applications. Performing two successive strong readout pulses separated by a time $\gg 1/\kappa$ would have clarified everything.

We did that and the result is shown in the two figures mentioned above. The highest obtained fidelity and QND-ness agrees with the previous data and analysis.

Below are further details of the new measurement the reviewer asked for.

We used two 500 ns long readout pulses separated by 1 microsecond to make sure they are not correlated and the falling tail of the first pulse does not affect the second one. We improved the window filter imposed by the downconversion frequency to 2 MHz and increased the cutoff frequency of the Chebyshev filter to 4 MHz. This settings ensures that the two readout pulses are uncorrelated. We collected 500 traces each for the qubit prepared in the ground and excited state and plot the results in an IQ plane for the first and second pulse to extract the threshold for recognizing ground and excited for different measurement powers. By calculating the probabilities as explained in the main text for QND-ness we extracted the following figure.

Where the IQ histogram for points a, b and c are the following:

The QND-ness at point a is limited by both SNR and qubit decay, at point b the states are well separated which lead to highest value of the QND-ness of 92% but still limited to the decay of the qubit and point c shows a non-QND measurement where the ground and excited state are mixed and a leakage to another state becomes visible.

Increasing the measurement speed by increasing window filter to 4 MHz and Chebyshev cut off to 8 MHz and reducing the pulses length to 250ns allows to perform the experiment at higher measurement powers but the ring down of the first pulse limits the start of the second pulse to about 750 ns and therefore the QNDness does not improve further than the number reported above.

The results in Fig. 7 have been updated to the results obtained by reviewer's suggestion and the main text changed accordingly.

- The authors mention multiple times the use of a Chebyshev filter applied to the measurement records. I do not understand why the author cannot show the raw data. In the literature, measurement records such as the one presented in Fig 7 a and b are shown without any additional filtering beyond the one of the detection chain (see e.g. Hatridge Science 2013 or Dassonneville PRX 2020). Could the author consider removing this filter ? It raises a lot of questions

We understand that this detail may have led to confusion. It is partially a result of the fact that we have developed a fast and efficient IQ receiver in house. Using a Chebyshev filter allows us to choose the resolution bandwidth of the measurement independent of the chosen analog IF frequency in a single channel acquisition. We emphasize that every measurement with any such instrument or software relies on such a filter that defines the resolution bandwidth no matter if the user realizes this or not. This is raw data (like a VNA or a SA we perform the filter and down-conversion in real time with C++).

So, providing this information was an attempt to full transparency. Nevertheless, the confusion is also understandable since we kept the oversampled data points and we used long pulses and narrow band filter with short integration times, which did not make a lot of sense and was misleading in fact.

The new data sets are based on short pulses and we remove details about filtering, or temporal mode matching since it really does not make a real difference for these results. Everything we show is correctly set up for 500 ns time resolution now. We can and we have performed these measurements also without the Chebyshev filter and for the specific measurements presented in this paper nothing changes given the IF frequency and thus the window filter are chosen correctly (and analog filters are used to avoid aliasing). Furthermore, we did not apply any temporal mode matching functions in the new version of the manuscript.

(i) Within a single trace, I imagine that successive measurement records for each time bin are now correlated since $1/1\text{MHz}$ is not negligible compared to the binning time of the measurement records shown in Fig 7a,b. What is the smoothing effect on the 'quantum jump' in b ? Claiming that this trace can be interpreted as a quantum jump after applying such a filter is non trivial to me.

In Fig. 13 b we show new quantum jump trajectories acquired with 500 ns time resolution and no oversampling.

(ii) What is the impact on the noise and the associated error bars for the mean measurement record ?

The figure and the method have been updated.

(iii) What is the impact on the 128 ns-long measurement used to determine the fidelity ?

The figure and the method have been updated. Even though factually correct this integration time was not the relevant number to report since the correlation length was significantly longer after applying the filter. We thank the referee for pointing this out.

(iv) What is the order of the filter ?

We use a Dolph-Chebyshev window FIR type filter implemented via Intel Integrated Performance Primitives that looks like this (dashed line is at 4 MHz):

It's based on a convolution with a kernel of length $2 \cdot f_s / f_{bw} - 1$ with the sampling frequency $f_s = 1$ GHz and the cutoff bandwidth of 4 MHz, i.e. $2 \cdot 250 - 1$.

- "a temporal mode matching function [...] was used".

I imagine that the data were processed further for the fidelity measurement. Not explaining the data processing (i.e. which function was used, why or the order of the filter) makes the results not very meaningful and irreproducible for other research groups. I am not even able to guess the function since different research groups use different filters. Is it a boxcar linear filter, an exponentially decaying linear filter, "optimal" linear/nonlinear filter, Bayesian filter ? That is critical information when discussing readout performances.

An exponentially decaying mode matching filter was used. However, we removed it altogether. Now we effectively apply a matched window filter (given by the IF frequency) and a Cheby filter followed by down-sampling i.e. keeping only 1 of the initially acquired 500 points.

- "While these results generally agree with the expectations from steady state simulations presented in [19]".

Similarly to the previous version of the paper, the authors want to make connections which are not supported by their results. I don't know why they want or need to say that. The paper by Verney et al. described the simulation of a circuit with parameters that are more than one order of magnitude different from the ones of the authors. For instance in their work $EL = 0.5$ GHz, while for Verney et al. $EL = 14$ GHz. I did not find in the work of Verney et al. any discussion about QNDnss or readout fidelity that can show an agreement with the results reported by the authors.

Our main point is that a QND readout is not possible with hundreds (or close to 1000) of photons in a transmon. At least to our knowledge. In our view the important difference is the potential confinement – even though very shallow in the present case – caused by the inductive shunt. In contrast to the transmon an RF-SQUID does not host states that are not bounded by the potential

(in contrast to the transmon). This is what we believe allows us to go beyond the photon assisted tunneling resonance conditions (see answers to referee 1) limiting the transmon.

While we understand that the simulations done by Verney, et al. have been conducted in a different range of parameters, we both promote the idea that the inductive shunt is beneficial to be able to use higher readout powers.

Now the claim is clearly there, I am therefore forced to ask the author to show an agreement between their Figure 7 and simulations such as the ones of Verney et al. in the relevant parameter regime of the experiment. In this way, one can see if there is an agreement between the simulation and the value of fidelity, QNDness, and typical photon number reported in Figure 7. Or at least one would be able to judge if the interpretation of Verney et al. is still valid for such a small inductive confinement.

We have changed the relevant sentence to: “These results support the intuition that an inductive shunt can suppress leakage to non-computational states due to the absence of unbounded states above the cosine potential as is the case for the transmon. A related effect has been shown in Ref.~\cite{verney_strongly_2018} in the limit of a significantly steeper parabolic confinement. Nevertheless, more detailed simulations of the strongly driven IST - resonator system, similar to the ones presented in Ref.~\cite{shillito_dynamics_2022}, are needed to better understand and utilize the IST qubit in the high-power limit.”

- The active fluxon excited state preparation is a very good idea. But in my understanding, the excitation scheme is not deterministic as stated in the conclusion?

We employ an active state preparation scheme as shown in Fig. 8b that always results in the desired state (repeat until success). The success probability of each individual attempt depends on the flux bias. Close to zero flux it is around 97%. Close to half flux it is around 50%. **More explanation is available in the new Methods section as well as Figure 14.**

Did the author consider a bichromatic initialization scheme ?

Considering a potential barrier as high as 60 GHz (2 EJ) and the plasmon energy of approximately 6 GHz, the first state with a wave function extending over two wells (the ancilla level) is above 20 GHz in frequency. We have not tried to excite the fluxons via such a state with our current control setup.

What is the success rate of each preparation cycle?

As explained in the main text using active state preparation scheme, the preparation of an excited state is guaranteed. Without the scheme and by calibrating the power where the excitation occurs we can reach an error rate as low as 3% near zero flux while near half of flux quanta this error increases.

What is the average duration of the total preparation?

The active preparation scheme can take multiple iterations to succeed, each iteration currently takes 100 ms set by the repeated plasmon readout, therefore the preparation cycle may take up to a few hundred ms. **This is shown in Fig 8b.**

- The authors claim a “perfect state preparation fidelity” of the fluxon with the protocol depicted in Fig. 8b. This must mean that the conditional plasmon excitation as well as its readout (in the single-photon limit) have a 100% fidelity ? That sounds too good to be true ?

Similarly, it would be good to explain how the finite plasmon measurement fidelity is taken into account to calibrate the y-axis of the Fig. 8c. Now it seems that it starts perfectly in 1 and ends perfectly in 0, which is only possible with the perfect state preparation fidelity mentioned above ? And the quantity $\langle \rho_{f01} \rangle$ reported on the y-axis is an element of a matrix density, I guess. Is it defined somewhere ?

We changed the wording and now avoid the word “perfect”. Nevertheless, the readout and thus also the active state preparation fidelity it is not limited by the single shot fidelity reported earlier since the fluxon preparation and readout is now based on an averaged readout relying on many plasmon preparations and readouts right after a single fluxon preparation (see also our answers to referees 1 and 2). The ultra-long fluxon lifetimes allow to essentially measure for a very long time (or many times) to acquire the required information (in which state are you?) with almost arbitrary precision (we only measure every 30 seconds after a successful excited fluxon state preparation), i.e. each state of interest is characterized by a specific dispersive resonator shift. Measuring the frequency (or the corresponding amplitude for a fixed frequency measurement) allows to distinguish these states and this calibrates the y axis. Here the plasmon state excitation is only used to boost the dispersive shift of the cavity when a fluxon tunneling event occurs, therefore increasing the SNR and making the f_0 and f_1 more distinguishable (this is most important near half flux).

The methods section contains now a lot more details including the pump power dependence for different flux bias values in Fig. 14.

- Could the author comment on the different discretization of the curves in Fig. 8c.? Is it due to a reduction of the dispersive shift between the plasmon and the resonator or between the fluxon and the plasmon ?

In each case we defined a threshold based on the dispersive shift/amplitude of the relevant states and the plotted curve is then a result of all measured fluxon jump events. The different discretization in x axis of the Fig. 8c is due to the fact that we have 80 tunneling events distributed in 25000 sec near half flux where the decay is slow in comparison to 8000 sec for zero flux where the decay is faster.

- The authors show impressive lifetimes for the fluxon transition. How to exclude other slow relaxation mechanisms such as the slow environment time relaxation reported <https://arxiv.org/pdf/2204.00499.pdf> ?

One specific answer we can give here is that we never observed a quantum jump back up to the excited fluxon state, i.e. without the resonator pump tone close to the resonance condition. Our data also shows that the system fully relaxes to the fluxon ground state for sufficiently long time scales. We would say that this excludes to a certain extent artificially longer lifetimes due to excitations from the environment. A related point was also raised by the reviewer 1 and to avoid repetition we refer the reviewer to point 9 raised by the first reviewer.

How can the reader be sure that the curves reported in Fig 8c are not due to slow temperature drifts of the measurement setup?

The base temperature of the fridge on average corresponds to 400 MHz in energy while the barrier between the wells is 60 GHz. The thermal relaxation rate scales exponentially with the ratio of the

height of the barrier and the thermal energy ($\hbar\omega_e - V_{\text{eff}}/k_B T$) and seems to be a negligible contribution to the quantum tunneling rate in the current sample. In the future biasing the qubit near half flux where the quantum tunneling rate starts to diverge and sweeping the fridge temperature can clearly identify limitations imposed by thermal fluctuations.

Are the lifetimes reported here compatible with the wavefunction overlap at different flux points? No they are not. SCQubits calculates 10^{21} seconds at zero temperature. This implies at least two things. First, we do not need to go to such a heavy limit to see similar timescales. Also, there are interesting effects to be explored in the future to better understand the physics limiting the observed fluxon lifetimes.

- “based on a 1 MHz Chebyshev filter”. I don’t understand the word “based” here, is it the correct word ?

This sentence was removed in this revision.

- “populate 40×10^3 data points in the quadrature histograms” -> is the word “populate” the correct word here ?

We changed the sentence to : “... we repeat the measurement 40×10^3 times to collect the data points for extracting the quadrature histograms ”

REVIEWERS' COMMENTS

Reviewer #1 (Remarks to the Author):

In this revision, the authors have compactified the manuscript by moving some technical details to Methods, which improves the readability of the work. Instead of directly addressing my concerns regarding the fluxon state readout/reset in my previous report, they argue that:

1. They will allocate a major effort to understand the non-QNDness and possible tunneling effects of the readout in a future paper.
2. It will take more time and effort to corroborate the fluxon lifetime data. This will possibly be published in the future.

Generally speaking, although unhappy about not seeing concrete explanations, I am satisfied with this approach. In my perspective, they report an intriguing observation, which may contain some interesting and unexplored dynamics. Currently, this is not well understood, despite a number of similar measurements, so I expect their results to provide additional motivation to investigate this phenomenon.

As for the fluxon's lifetime measurement, the manuscript transparently describes the methodologies and thoroughly discusses the results. This should allow other researchers to measure the IST and check for consistency if needed. As the field is still evolving rapidly, possible discrepancies in the future can be attributed to the limitation in metrology.

I still see some minor problems with the writing, which I expect to be smoothed out during production. I can recommend this work for publication.

Reviewer #2 (Remarks to the Author):

The revised manuscript has satisfactorily addressed my previous concerns. The added section and modifications in the latest version have considerably improved the quality of the paper and I recommend publishing in Nature Communications.

Reviewer #3 (Remarks to the Author):

I would like to thank the authors for replying thoroughly to the referees' comment. I think that the review process substantially strengthened and clarified the paper. I hope that the authors agree with this statement. The authors managed to provide insightful answers to all the questions during the last review round.

The section of the paper dealing with readout fidelity and QNDness is now very rigorous compared to the last version. This work will definitely be very valuable to fully elucidate the subtle behavior of the fluxonium-like circuits at very high power. The impressive T_1 of the fluxon transition will likely trigger a lot of attention in quantum physics as it is both a fundamental test of Fermi's golden rule and also a stress test of the "no wavefunction overlap" strategy followed by many research groups working on protected qubits. Finally, the IST is a new parameter regime that will be potentially advantageous over other circuits to manipulate, store and read out quantum information in the context of superconducting quantum computers.

I could not spot any remaining issues or any typo in the paper.

For all these reasons, I would like to warmly recommend the article by F. Hassani et al. for publication in Nature Communications.